# A genomic toolkit for winged bean *Psophocarpus tetragonolobus*

Wai Kuan Ho [1,2,7], Alberto Stefano Tanzi[3,7], Fei Sang [4,7], Niki Tsoutsoura [3,7], Niraj Shah [5,7], Christopher Moore [4], Rahul Bhosale [3], Victoria Wright[4], Festo Massawe [1] & Sean Mayes [2,3,6] ✉

A sustainable supply of plant protein is critical for future generations and needs to be achieved while reducing green house gas emissions from agriculture and increasing agricultural resilience in the face of climate volatility. Agricultural diversification with more nutrient-rich and stress tolerant crops could provide the solution. However, this is often hampered by the limited availability of genomic resources and the lack of understanding of the genetic structure of breeding germplasm and the inheritance of important traits. One such crop with potential is winged bean (*Psophocarpus tetragonolobus*), a high seed protein tropical legume which has been termed 'the soybean for the tropics'. Here, we present a chromosome level winged bean genome assembly, an investigation of the genetic diversity of 130 worldwide accessions, together with two linked genetic maps and a trait QTL analysis (and expression studies) for regions of the genome with desirable ideotype traits for breeding, namely architecture, protein content and phytonutrients.

Winged bean (*Psophocarpus tetragonolobus* (L.) DC.), also known as goa bean, is a dicotyledonous ($2n = 2x = 18$) legume belonging to the Fabaceae family and Papilionoideae subfamily. It is the only domesticated species of genus *Psophocarpus* and has three pairs of short and six pairs of long chromosomes[1]. It is cultivated mainly in the hot, humid, equatorial countries of Southern Asia, Melanesia and the Pacific region. It is grown as a vegetable for its immature pods, and for leaves and tuberous roots, with consumption preferences differing geographically[2]. Its ability to fix atmospheric nitrogen with a wide range of Rhizobium[3,4] makes it an ideal horticultural crop for more sustainable agriculture particularly in the humid and sub-humid tropics. Nevertheless, its uncontrolled branching habit, indeterminate growth and inconsistent productivity all limit current large-scale utilisation. Our previous work has suggested that an optimal trade-off could be achieved between vegetative growth and yield (pod number and seed number), for example, by controlling branch number[5].

Individuals with fewer but longer branches could maintain high pod productivity, while potentially reducing the vegetative biomass.

Winged bean's description as 'a soybean of the tropics' was coined in the 1980s after work by the US National Academy suggested it had potential to partially replace soybean in tropical countries[6]. In particular, the mature seeds are a good source of protein and other nutrients. Mature dried winged bean seeds are reported to contain 16.4 to 21.3% lipid and 34.3 to 40.7% crude protein with a favourable amino acid balance and micronutrient composition[7,8]. However, its use for dietary diversification and nutritional potentials have been hampered by the crop architecture and the 'hard-to-cook' phenomenon, while the presence of anti-nutritional factors, such as hemagglutinating lectins, trypsin and chymotrypsin inhibitors, and tannins in the seed represent a challenge to protein digestibility and nutrient bioavailability[9]. The crude protein content in the seeds of 25 accessions from the International Institute of Tropical Agriculture (IITA) ranged from 28.4 to 31.3%

[1]Future Food Beacon, School of Biosciences, University of Nottingham Malaysia, Jalan Broga, 43500 Semenyih, Selangor, Malaysia. [2]Crops for the Future (UK) CIC, NIAB, 93 Lawrence Weaver Road, Cambridge CB3 OLE, UK. [3]Future Food Beacon, University of Nottingham, Sutton Bonington Campus, Loughborough, Leicestershire LE12 5RD, UK. [4]Deep Seq, Centre for Genetics and Genomics, University of Nottingham, Queen's Medical Centre, Nottingham NG7 2UH, UK. [5]Digital and Technology Services, University of Nottingham, Sutton Bonington Campus, Loughborough, Leicestershire LE12 5RD, UK. [6]International Centre for Research in the Semi-Arid Tropics (ICRISAT), Patancheru, Hyderabad 502324, India. [7]These authors contributed equally: Wai Kuan Ho, Alberto Stefano Tanzi, Fei Sang, Niki Tsoutsoura, Niraj Shah. ✉e-mail: Sean.Mayes@nottingham.ac.uk

with tannin and phytate varying significantly[10]. Some of the issues caused by antinutritional factors can be overcome through different cooking and processing methods[10,11]. Coupled with architectural changes and breeding selection for high protein and/or high lipid, genetic improvement could widen winged bean production from a purely horticultural crop to an arable crop for seed protein, also helping to reduce dependency on soybean imports for many countries and the environmental impact associated with the expansion of soybean production, especially in South America.

Genome-assisted breeding improvement strategies could lead to a genetically improved winged bean that contributes to food and nutritional security. However, the lack of molecular resources has hampered a better understanding of the diversity and phylogenetic relationships of germplasm conserved ex situ. To address this knowledge and resource gap, here we generate a chromosome-level genome assembly through a combinatorial approach integrating Illumina and Nanopore reads with Bionano optical mapping and within species genetic mapping. Using Genotype-by-Sequencing (GbS), we also evaluate the genetic structure of a wide range of global accessions, develop two controlled cross genetic maps and perform a quantitative trait loci (QTL) analysis of architectural and nutritional traits.

## Results

### Genome assembly and annotation

Ma3, a cultivar (renamed from the previously reported 'M3'[5]) released by the Malaysian Agricultural Research and Development Institute (MARDI) was selected for genome sequencing. Our approach of using Oxford Nanopore Technology (ONT) reads polished by Illumina reads generated 1224 contigs with the largest being 30.731 Mb in length. An improvement was further made with the addition of Bionano optical mapping, yielding a total of 46 hybrid scaffolds with the largest scaffold being 42.435 Mb and a N50 of 28.31 Mb (Table 1 and Supplementary Data 1). A total of 96.8% of Illumina reads were successfully remapped back to the assembly.

Subsequently, SNP markers in two genetic maps derived from Genotype-by-Sequencing (DArTSeq™) were used to assign scaffolds for the final assembly at a pseudo-chromosome level, spanning

586.44 Mb in total. This assembly accounts for 98.9% of the assembled scaffolds, with only 6.41 Mb of scaffolds left unassigned. The spaced marker genetic maps derived from Cross XT ($F_2$ = 184) and Cross XB ($F_2$ = 223) have Ma3 as the common paternal parent (Fig. 1a). The genetic map of Cross XT comprises 692 high quality, spaced, SNP markers covering 1102.1 cM (an average interval of 1.6 cM) while the XB map comprises 393 spaced SNP markers covering 1336.9 cM (3.4 cM average distance per marker). Both maps were constructed to have no missing data across the whole population. (Supplementary Data 2). The nine anchored pseudo-chromosomes ranged from 27.50 to 91.98 Mb in length (Supplementary Data 2). Of the SNP markers present in the linkage maps, 98.1% (XT) and 96.2% SNPs (XB), could be mapped back to the genome assembly, respectively. From the full DArTseq™ dataset derived from the mapping and diversity analysis, 96.9% of SNP ($n$ = 23,134) and 79.2% ($n$ = 42,773) of presence-absence variants (PAV/in silico DArT), respectively, had their sequence tags mapped back to the final genome (Fig. 1b). Repetitive elements represent 58.95% ($n$ = 881,111) of the winged bean genome with the majority (in terms of number) being Terminal Inverted Repeat elements (TIR; 58.50%, 149,593,269 bp) followed by Long Terminal Repeat elements (LTR; 27.71%, 147,429,990 bp) (Supplementary Fig. 1a, b). The Class I/Class II transposable element (TE) number ratio was observed to be 0.55 (Supplementary Data 3–4). Of the dominant LTR in Class I retrotransposons, the *Gypsy* superfamily was 3.8 times more abundant than *Copia*-types.

Other than de novo gene prediction and homology comparison, previous transcriptomic resources from leaf, root, pod, and reproductive tissue (comprising of bud and flower)[12] were also included, assisting in the annotation of 30,397 protein coding genes. Among these, 13,139 were assigned with GO terms, 11,898 with KO terms and 7390 with KEGG pathway positions by eggNOG-mapper. Of 1614 Embryophyta larger group single-copy core genes from Benchmarking Universal Single-Copy Orthologs (BUSCO) analysis, 96.2% complete conserved genes were recovered with 2.7% as missing and 1.1 % fragmented (Table 1). Additionally, approximately 85% of the annotated genes have an Annotation Edit Distance score ≤0.3[13] (Supplementary Fig. 2a), indicating a high degree of quality and continuity for the current annotation. These gene models have an average length of 4254 bp and an average coding-sequence length of 1324 bp (Supplementary Fig. 2b–d), with an average of five exons per gene (Supplementary Fig. 2e). As many as 1641 of the predicted genes are transcription factors, accounting for 5.4% of the total annotated genes. Together, transcription factor classes MYB and bHLH are dominant and accounted for 19.68% of the identified transcription factors (Supplementary Table 1). Additionally, 14,852 small nucleolar RNAs and 1044 tRNA genes were identified.

The previously reported genome size of winged bean varies widely; Vatanparast and colleagues have suggested it to be 1.22 Gbp/1C in size whereas previous flow cytometry has estimated it to be 782 Mb1C[14,15]. Our *K*-mer distribution analysis predicated a smaller genome size at 569 Mb, although GC-rich regions and repetitive sequences might be underestimated (Supplementary Fig. 3). In short, the current reference genome 'covers' 103% of the *K*-mer estimated genome size of 569.1 Mb and thus, is expected to provide comprehensive coverage of the coding regions of winged bean.

Dispersed duplication ($n$ = 16,452, 58.69%) was the predominant type of gene duplication in winged bean, followed by 4709 (16.8%) of whole genome /segmental duplication and 15.72% ($n$ = 4407) transposed duplicated genes. Tandem duplication and proximal pairs constitute the least, at 4.70% ($n$ = 1317) and 4.10% ($n$ = 1148), respectively (Supplementary Data 5). The longest stretches of chromosomal collinearity are between Chr01.1 (17.39 Mb) and Chr03.5 (11.92 Mb), comprising of 244 genes (Supplementary Fig. 4).

The evolutionary comparison of winged bean gene models with nine other leguminous species suggests that 5.4% of the genes are

**Table 1 | Statistics summary of reads and assembly**

| Pseudochromosome, integrated with genetic maps | |
|---|---|
| Scaffolds assigned to LGs | |
| Cross XB (FP15 x Ma3) | 34 |
| Cross XT (Tpt10 x Ma3) | 36 |
| Unassigned scaffolds to LGs | 10 |
| Length of assigned sequences to LGs | 580,032,372 bp |
| Length of unassigned sequences to LGs | 6,405,169 bp |
| Min scaffold length (Mbp) | 0.096 |
| Median scaffold length (Mbp) | 8.078 |
| Mean scaffold length (Mbp) | 12.749 |
| N50 scaffold length (Mbp) | 28.310 |
| N50 contig length (Mbp) | 10.861 |
| Max scaffold length (Mbp) | 42.435 |
| Total scaffold length (Mbp) | 586.438 |
| Repeat region of assembly (bp) | 202,036,684 |
| Completeness of gene prediction (BUSCO) | |
| Complete genes | 96.2% |
| Complete and single copy | 93.8% |
| Complete and duplicated | 2.4% |
| Fragmented genes | 1.1% |
| Missing genes | 2.7% |
| Total BUSCO groups searched | 1614 |

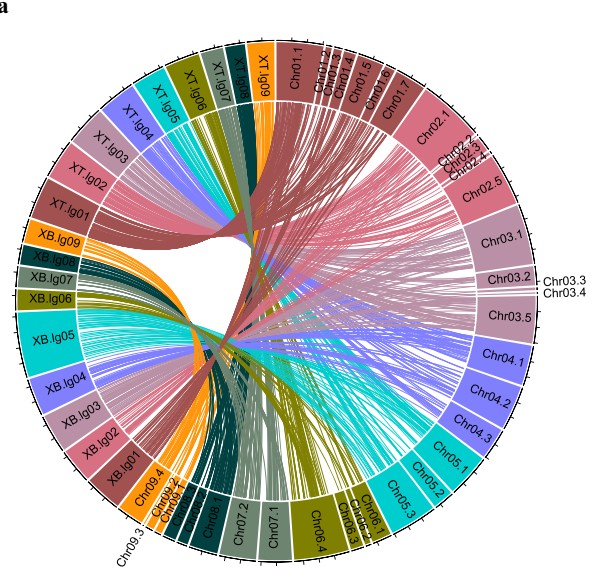
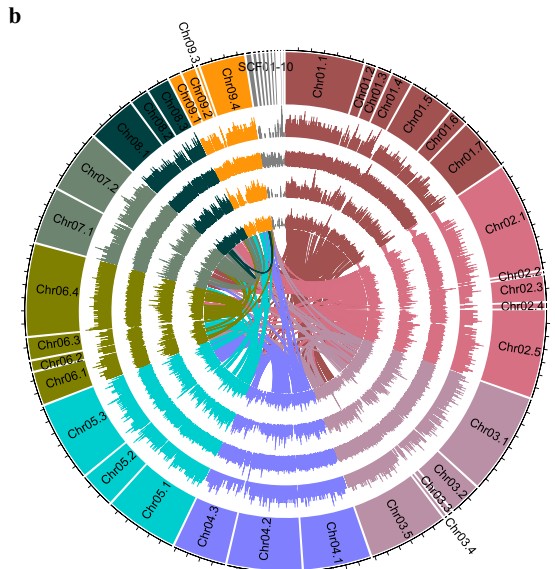

**Fig. 1 | Overview of the winged bean [*Psophocarpus tetragonolobus* (L.) DC] genome assembly. a** anchored by two genetic maps (Cross XT and Cross XB) and its (**b**) genome characterisation. The blocks represent nine pseudomolecules featured in 5 Mbp intervals. Tracks in (**b**) displayed are as follow: outer most (i) chromosome name, (ii) gene density, (iii) TE elements density, (iv) SNP density, (v) presence-absence variation density (PAV), (vi) intra-genomic syntenic blocks. The density was calculated using 10 kb non-overlap window.

species-specific while 25,471 genes (83.8%) belong to 16,356 gene families. A total of 11,263 orthogroups are found with 635 of them single-copy genes (Supplementary Data 6). Interestingly, winged bean is most closely related to *Glycine* rather than other legume species and is deduced to have diverged approximately 14–16 Mya, before *Glycine* doubled its chromosome number (Fig. 2a). The estimated synonymous substitution rate (*Ks*) distribution of the winged bean paranome reveals that it has undergone the legume-common tetraploidy duplication event (Fig. 2b) in Fabaceae as in other legumes[16]. Consistent with this, winged bean shares 58.3% and 55.3% collinearity with *G. max* and *G. soja*, respectively, with the largest collinearity blocks observed between Pt04 and Gm10 (Fig. 2c and Supplementary Fig. 5). Nevertheless, extensive chromosomal rearrangement or gene translocation can be observed between them (Supplementary Fig. 5a, b). Winged bean has not only the smallest gene family expansion (*n* = 221) but also the highest gene family contraction (*n* = 1490) among these legume species (Fig. 2a). Interestingly, the winged bean genome appears enriched for genes involved in isoflavonoid biosynthesis and secondary metabolites (such as phytosterol, castasterone and ergocalciferol), potentially functioning as phytoalexins through jasmonate- and ethylene-mediated pathogen defence mechanisms (Supplementary Fig. 6)[16].

### Winged bean germplasm diversity

The genetic diversity of winged bean germplasm was investigated using 168 individuals originating from 17 countries (Supplementary Data 7) based on GbS (DArTSeq) markers. Using 10,523 SNPs it was observed that the level of heterozygosity ranged from 1.03 to 26.33% with a median value of 5.04% per genotype, largely reflective of a self-pollinating species (Supplementary Data 7). Due to the selection nature of the DArTSeq™ approach in targeting single copy sequences and gene-rich regions (59.1% are genic SNP), it is perhaps not surprising that the technique is able to reveal high individual genotype heterozygosity rates in some accessions[17,18]. Nevertheless, the presence of individuals with high heterozygosity implies outcrossing and is consistent with our observations of bees and other insects visiting the large flowers in open field conditions. An 16.4% average individual heterozygosity has been reported from 457 Thai accessions evaluated using 14 Simple Sequence Repeat markers[19].

Bayesian clustering and phylogenetic analysis [Neighbour-Joining (NJ) tree] and Principal Coordinate Analysis (PCoA) results all closely corresponded to each other and suggest the existence of four major gene pools (Fig. 3a–c, Supplementary Fig. 7a). Based on a membership coefficient of ≥0.7 with approximately 20% of these lineages showing allelic admixture. As previously reported[2,20,21], the clustering correlates, to a limited extent, with the country of origin, yet the possibility of seed exchange among genebanks (with limited documentation or historical exchange with subsequent collection capturing the secondary collection sites) cannot be discounted (Supplementary Fig. 7b, c). Comparing among countries with more than five accessions in this study, 37.5% and 28.6% of the materials originating from Colombia and Nigeria show high levels of admixture. This suggests possible in-field hybridisation events between accessions, from genetic exchanges of germplasm. This further supports the hypothesis that germplasm was most likely introduced from Asia to South America, given that winged bean is not reported as a native or traditional crop consumed in South America[2]. For 54 accessions, co-accession samples from the same breeding line based on pedigree information were available. However, 30% (16 out of 54) of these were found not to be highly similar when evaluated by DArTseq SNPs (Supplementary Fig. 7d). Although mislabelling at some stage cannot be ruled out, outcrossing might have also contributed to the differences between genotypes, or it may reflect incomplete inbreeding at the stage at which pairs were sampled from the breeding programme. As such, our findings could help breeders by emphasising the need for quality assurance for breeding programmes and to test pedigree information, as well as providing insights into some of the publicly available germplasm.

Analysis of Molecular Variance (AMOVA) reveals that the genetic variance observed is not only attributed to the differences between individuals within population (49.2%) but also within individuals (31.4%), resulting in population differentiation (Supplementary Table 2). The first principal component (Coordinate 1 = 37.0% variation explained) in the PCoA distinguishes most of the Thailand materials in

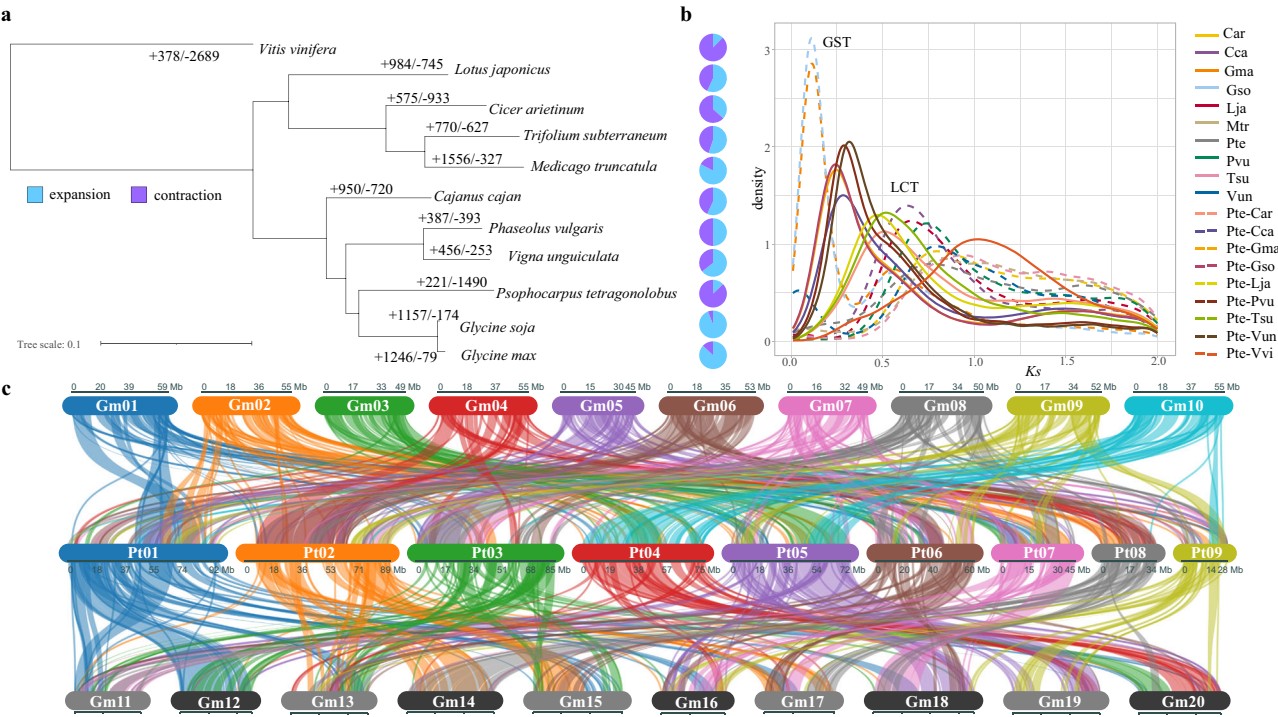

**Fig. 2 | Evolutionary analysis of winged bean with nine legume species.**
**a** Proportion of expanded (blue) and contracted (purple) gene families in ten legume species. **b** *Ks* plot of the paralogues and orthologous genes of ten legumes showing the glycine-specific tetraploidy (GST) event in *G. max* and *G. soja* and the legume-common tetraploidy (LCT) event experienced in all legumes. Car: *Cicer arietinum*, Cca: *Cajanus cajan*, Gma: *Glycine max*, Gso: *Glycine soja*, Lja: *Lotus* *japonicus*, Mtr: *Medicago truncatula*, Pte: *Psophocarpus tetragonolobus*, Pvu: *Phaseolus vulgaris*, Tsu: *Trifolium subterraneum*, Vun: *Vigna unguiculata*. **c** Conserved syntenic blocks between winged bean and *G. max*. A block of 1000 Ns was added between super-scaffolds within same chromosome for chromosomal level visualisation.

Subpopulation Q1 from Indonesian material in the Subpopulation Q2. The differentiation of Q3 and Q4 subpopulations from the others is revealed mainly by the second component (Coordinate 2 = 22.0% variation explained). While moderate, the greatest genetic divergence (0.4461) was observed between the Q1 and Q3 subpopulations, in line with the population structure analysis revealing that Q1 accessions are clearly separated into a single cluster from $K = 2$ onwards (Supplementary Table 3). Being the smallest cluster, its narrow genetic base makes it distinctive when $K = 3$ (Supplementary Data 8). As the likely donor of the Colombian materials evaluated in this study, the late Professor Theodore Hymowitz documented in 1977 that Papua New Guinea is likely to be the centre of origin for winged bean[22], implying that at least part of the Colombian accession base was introduced from Papua New Guinea. The Q4 cluster predominantly consists of accessions collected from the Philippines and is closely related to the Q2 gene pools, in line with the geographical status whereby approximately 30% of the accessions in this cluster are Indonesian accessions. Overall, observed heterozygosity was much lower than the expected heterozygosity in all gene pools with the Q2 cluster being the most diverse ($H_e = 0.209$), in agreement with the broadly inbreeding nature of winged bean.

### Linkage disequilibrium (LD) decay
The genome-wide linkage disequilibrium decay rate of winged bean was estimated to be in the range of 262 to 494 kb with an average of 387 kb. A total of 10,534 SNPs were distributed across the genome with 1170 SNPs per chromosome on average, with Pt08 showing the fastest LD decay (Supplementary Fig. 8 and Supplementary Table 4). Only 3.4% of SNP pair showed high LD ($r^2 \geq 0.8$). This LD decay rate is similar to other closely related autogamous legumes such as soybean (250–375 kb)[23,24] and common bean (183–397 kb)[25,26].

### Quantitative trait locus analysis of XB population
As a first step in breeding and trait analysis we developed a number of controlled crosses with the Ma3 genotype as the common paternal parent, focusing on architecture and pigmentation. Two of these populations were used for genetic map construction to assist in scaffold assignment and orientation to generate the physical map of winged bean (above). The segregation for morphological traits in the XB $F_2$ between FP15 and Ma3 has previously been reported[5]. For this cross, a QTL analysis was also conducted (illustrated in Supplementary Fig. 9) and is summarised in Supplementary Data 9. To test the utility of the genome resources generated, we then investigated the inheritance of three traits of potential breeding interest in the XB population in additional to those previously reported[5].

### Variation in seed storage proteins (SSP) compared to soybean
The review by Kadam et al in the 1980s reported that winged bean seeds have a comparable essential amino acid composition to soybean[7]. From this base, molecular breeding could enhance protein composition and content. It was also reported that protein components in the winged bean seed are predominantly vicilin-type 7S globulin occupying approximately 31.9% of the protein extract, with a lack of 11S legumin-type[27,28]. The paternal Ma3 genotype had a measured average of 34.0 ± 1.8% ($n = 14$) total seed protein content, while the maternal FP15 had 39.3 ± 3.2% ($n = 9$), although wider differences between accessions have been observed between lines from other studies[6]. Among the $F_3$ seed from the XB cross (bearing in mind that the seed largely represents the maternal genotype; $n = 161$ lines evaluated), transgressive segregation was observed in the population, ranging 27.2 to 44.6% total protein per seed (Fig. 4a). In other legumes such as pigeonpea (*Cajanus cajan*), soybean and pea (*Pisum sativum*) similar seed protein segregation was reported, although

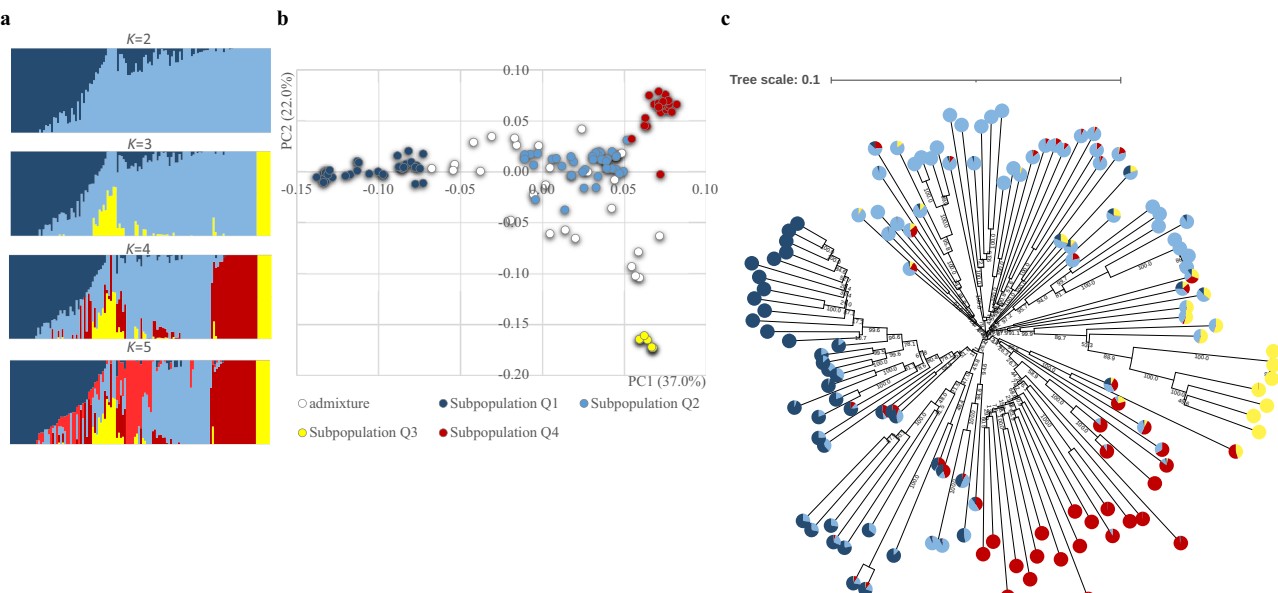

**Fig. 3 | Genetic divergence analysis of 130 germplasm accessions. a** Population structure membership probability plot from *K* = 2 to *K* = 5. **b** Clustering from PCoA analysis; suggesting the most likely number of subpopulations is four. **c** Phenetic relationship of germplasm included in this study with pie charts depicting subpopulations (Subpopulation Q1, Subpopulation Q2, Subpopulation Q3 and Subpopulation Q4).

24.6% in pigeonpea and 33.0% in pea were considered as high protein genotypes[29,30], although lower than seen here in winged bean.

In addition to the 23 genes putatively encoding 7S globulin identified from the genome sequence (Supplementary Data 10), we also found two tandem genes (Psote03G0078700 and Psote03G0078800) homologous to the 11S globulin in soybean, *Medicago truncatula* and chickpea, approximately 6.6 Mb from the *qSSP-1* QTL, which were not identified by previous studies. As legumin contains a higher proportion of sulphur-containing amino acids than 7S vicilin/convicilin[31], it is interesting to note that Psote03G0078700 was not only predominantly expressed (compared to its paralogue) in the pod tissues of the FP15 maternal line but had the second highest abundance among the identified seed storage protein genes at the early stage of pod development (Supplementary Data 11). Its expression declined as the pod matured while the transcripts of a putative albumin (Psote06G0099600) increased (Fig. 5 and Supplementary Data 11). 2S albumin is another protein component rich in sulphur in common bean[31] that varied among different winged bean cultivars[28]. It was reported to be the second most abundant protein fraction occupying 26.5% of total protein in winged bean. Given that one of the expanded gene families in winged bean is involved in sulphur compound biosynthesis (Supplementary Fig. 6), quantifying sulphur content in each protein fraction could provide further targets for winged bean genetic improvement for good nutritional quality and amino acid balance.

In contrast to soybean, seed developmental stage characterisation has suggested that protein bodies in winged bean seed accumulate at a late stage of maturation (45 days after flowering) when starch granules are decreasing rapidly[32,33]. Added to that, our data illustrated that the high protein FP15 genotype has already accumulated as much protein as the mature seed of Ma3 genotype at Day 37 after anthesis, with a significant further increment by Day 45 after anthesis when compared to Day 30 ($p < 0.005$, Fig. 4b). The protein profiles characterised by Makeri and colleagues further indicated that the N-content in the globulin fraction is highest in winged bean seed, but lowest in soybean globulins, but also that N-content is significantly different from other protein fractions in both crops[27]. These observations, in turn, might suggest that the targets for genetic improvement in seed protein content or quality could be different between winged bean and soybean.

A significant QTL explaining 10.8% of the protein variation was detected on Pt03.1. Five seed storage protein genes (one being putative) as well as two NAC transcription factor were identified within three of the four putative QTL, with the seed protein QTLs cumulatively explaining 26.8% of the total phenotypic variation (Supplementary Data 9). Whilst these genes were not highly expressed in the developing pod, missense changes were found in two of the genes between the parental lines, suggesting potential variations linked to protein accumulation differences (Fig. 5 and Supplementary Data 11). Given the high genetic complexity underlying the accumulation of seed protein - for example, 33 consensus amino acid content QTL were identified in soybean[34]—it is not surprising that candidate gene identification is not straightforward (Supplementary Data 12). As an example, *Psote03G0119900* is homologous to xylem bark cysteine peptidase 3 (P34, Glyma.08G116300), which had the second highest abundance detected in all four stages of soybean seeds during storage protein accumulation[35], despite its lower expression level in winged bean pods (Fig. 5). Although this is not a well characterised gene, it is highly expressed at 40 days after flowering (DAF) in soybean and associated with enriched GO terms 'transmembrane transport' and 'nutrient reservoir activity'[35]. In winged bean, it had a 9.5-fold up-regulation in the maturing pods (45 days after anthesis) as compared to 15 days of immature pods. Furthermore, using the same accessions, the authors compared their DEGs with the Harada-Goldberg soybean RNA-seq dataset (http://seedgenenetwork.net) which comprises of 15 tissues, with this gene found to be tissue-specific and expressed during early- to mid-filling seed stage (from stages s1 to s6, with increasing expression levels out of nine seed stages)[35]. Despite soybean Glyma.08G108800 (*PtSHM1*; Psote03G0113400) and Glyma.08G108800 (*PtSHM2*; Psote03G0113600 within *qSSP-1*) not being tissue specific, they were expressed highly in all four soybean seed developmental stages, a similar observation to that in the winged bean developing pods (Fig. 5 and Supplementary Data 12), both of which are involved in amino acid (glycine and L-serine) biosynthesis[35]. The high expression of *PtSHM1* and *PtSHM2* in the developing pods would suggest that they contribute to high protein content in winged bean pods, previously reported to range from 2.9 to 3.0% (wet weight) in immature and mature pods[36]. Despite the expression level of *PtSHM1* being nearly two-fold higher than *PtSHM2* in all pod stages, three indel sites totalling

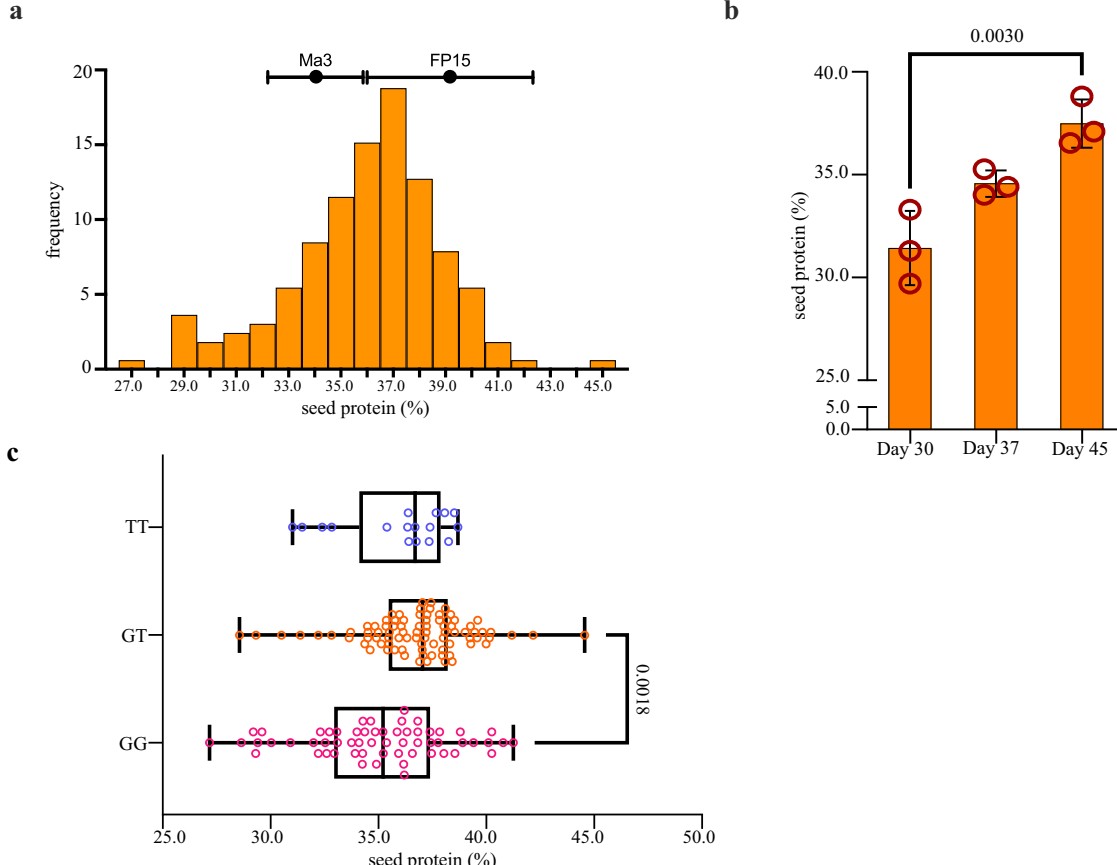

**Fig. 4 | Seed storage protein in the F₃ seed of XB population. a** Transgressive segregation in seed protein content (%) observed in the XB population ($n = 161$). **b** Bar plot showing progressive seed protein accumulation in the maturing seeds of FP15 maternal line, each with biological triplicate replicates ($n = 3$). Data are presented as mean ± SEM. Different letters indicate statistically significant differences at $p < 0.01$ by one-way ANOVA test with Tukey post-hoc test (Day 30 vs Day 45, $p = 0.0037$). **c** Median values of seed protein content in XB F₃ population ($n = 161$) were relatively higher (37.0% with few outliers observed) in the heterozygous lines

with SNP Marker_33026288 carrying a G/T SNP variation at Chr03.1: 23,735,683 than the GG (35.2%) and TT (36.7%) homozygotes. Box plot (drawn using R geom_boxplot) with the centre line indicates the means, the edges represent 25th and 75th percentile, the whiskers extend to indicate the 1.5 interquartile range from the edges. Different letters indicate statistically significant differences at $p < 0.001$ by one-way ANOVA test with Tukey post-hoc test (GT vs GG, $p = 0.0007$). Source data are provided in a Source data file.

17 bp at the 5′ UTR of *PtSHM2* potentially (Supplementary Data 11) could lead to superior seed protein levels in the progeny of these lines, as both parents are heterozygous in these regions. In spite of *PtNPF* (Psote03G0123600) being expressed at low levels (0.005–0.212 FPKM) throughout the pod maturing stages, a SNP marker at the intron of this gene could be an informative biomarker for seed storage protein as the heterozygous lines carrying G/T were on average 0.8% and 1.8% higher than T/T and G/G lines, respectively (Fig. 4c). Furthermore, a change of neutral Serine to hydrophilic Asparagine in Exon 5 (Ser536Asp) of this gene is observed in the FP15 material. Arabidopsis NPFs are involved in nitrate sensing and transport, particularly NPF2.12 and NPF5.5, to ensure nitrate supply and nitrogen accumulation in seeds[37]. Given that PtNPF has the highest similarity with AtNPF4.6 which is responsible for nitrate uptake in Arabidopsis roots, the tissue specificity of this gene in winged bean remains to be explored. Additionally, the DELLA protein *RGL3* orthologue (Psote07G0210200) located at *qSSP-5* with 8.0% PVE was highly and consistently expressed in maturing pods and coincides well with the role of *AtRGL3* being the coactivator of *AtABI3*, to mediate seed protein storage accumulation[38]. Overall, from this initial work it is clear that there is potential within winged bean germplasm to select for a high protein legume seed crop, either through conventional approaches or marker-assisted selection.

At the moment, winged bean is primarily a horticultural crop, grown on trellis structures and manually harvested. For large-scale seed production, the crop could be grown as a creeping vine on the flat (assuming that flowering and pod development is not interrupted) or changes in the architecture of the plant could be bred for, to facilitate mechanisation of sowing and harvesting or even to optimise it for intercropping with complementary amino acid sources, such as cereals.

## Altering architectural traits to improve mechanisation

Our previous work[5] suggested that fewer but longer branches could achieve the best trade-off between yield and vegetative biomass. Both *branch length* and *total number of branches* traits had a significant positive correlation with *pod number* ($r_s = 0.44$, $p < 0.001$ and $r_s = 0.28$; $p < 0.01$, respectively). Selection for these traits could therefore have a major impact on plant architecture and yield of winged bean. There could also have been selection for this trait during domestication, given the observed reduction in the higher orders of branches in the domesticated species, compared to closely related species like *P. scandens*. Our QTL analysis has further suggested a putative *qNoB* (*branch number* QTL, measured from branches >10 cm in length within the first ten nodes) that co-localises with a significant *total branch number* QTL (*qTNoB-1*; LOD 4.4, 21.1% PVE, counted from the entire main stem) with the former having a narrower 2-LOD confidence interval (Chr03.5: 14,853,644. 26,464,617), implying that yield variation in this material may be contributed to by branches departing from the

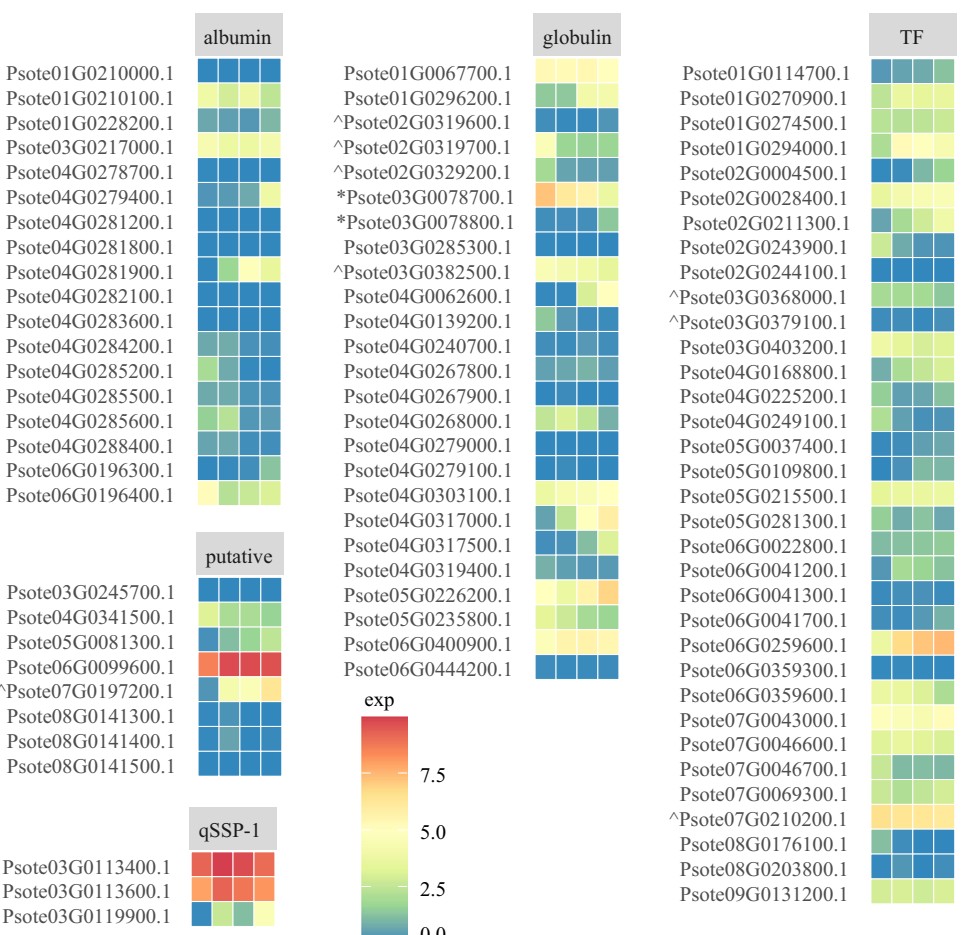

**Fig. 5 | Expression profile of seed storage protein related genes in developing pod tissues.** Expression levels [log$_2$ (FPKM+1)] of albumin, globulin, associated transcription factors and related genes (listed in Supplementary Data 10) in the pod tissues at Day 15, Day 30, Day 37 and Day 45 after anthesis (from left to right). '*' indicates genes near *qSSP-1* and '^' indicates genes within *qSSP-2*, *qSSP4* and *qSSP-5*. Source data are provided in a Source data file.

upper stem nodes. Although lower *branch number* but longer *branch length* could be a better selection criterion for yield, the *branch length* QTL (*qLoB*) is currently putative at LOD 3.6. CUC2 is a member of the NAC domain of transcription factors that defines organ boundaries and organ numbers[39]. As both the At*CUC2* orthologues (Psote01G0148300 and Psote03G0327500) are located within the QTL (*qTNoB-1*/*qNoB* and *qTNoB-2*), they are promising candidate genes given observed changes in the protein around the conserved NAM domains due to missense mutations in *PtCUC2a* (four nonsynonymous mutations) and *PtCUC2b* (an amino acid insertion and a nonsynonymous change) in the FP15 material (Supplementary Data 13 and Supplementary Fig. 10). Li and colleagues have recently reported the CUC2/3-DA1 UBP15 regulatory module and its role in controlling axillary meristem initiation[40]. The *AtUBP15* orthologue (Psote03G0312000) is positioned within the *qTNoB-1*/*qNoB* region as well.

**Inheritance of phytonutrient content - purple pigmentation**
The maternal parent of the XB population (FP15) has pods which are near-full or full purple when they approach maturity, while the seeds are purple or near black (Supplementary Fig. 11). FP15 is a member in the Q3 subpopulation with extremely low heterozygosity, perhaps suggesting that this line might have undergone active selection by farmers for its distinctive pigmentation. The paternal parent Ma3 (Q2 gene pool) has green stems and fully green pods, with the F$_3$ cross population testa (F$_2$ maternal tissue) having pigment segregation in the pods, seed testa and for pod 'specks'. As the seed coat is a

maternal tissue, it is not surprising that F$_1$ seed were dark purple without segregation. Of the F$_3$ seed screened, 80 had a brown seed coat and 122 were purple (Supplementary Fig. 11). This ratio ($\chi^2 = 1.411$; $p = 0.24$) might be attributed to two independent dominant loci with a duplicate recessive epistasis interaction. Two major significant QTL (*P1* and *P2* alleles, Supplementary Data 9) were detected for these two traits explaining 61.7% and 58.5% of the phenotypic variation, respectively.

These pigmentation loci (Supplementary Data 9) demonstrate the complexity in the regulatory network of the anthocyanin biosynthesis pathway in a tissue-specific manner. The pigmentation in pod and calyx followed the same segregation patterns as the seed testa ($\chi^2 = 0.067$, $p = 0.80$; $\chi^2 = 0.229$, $p = 0.63$) and exhibited co-dominance and/or an incomplete dominance. Of these traits, the F$_1$ had purple-green calyx and purple specks on the pods. While the pod pigmentation varied in speck quantity and even extended to purple 'wings' (pod fringes), no F$_2$ progeny were identified which had as full a purple pod as the maternal parent. Notably, lines with green calyx had brown seed testa, suggesting the lack of anthocyanin expression in both tissues (Supplementary Fig. 12). In support of this notion, QTL for *calyx colour* and *pod pattern* were identified also on Pt02, Pt03 and Pt09, in addition to the *P1* and *P2* loci. In soybean, there have been six loci identified; *I*, *T*, *Wp*, *W1*, *R* and *O*, producing seed colours ranging from yellow, green buff and brown to black[41]. Both brown (*irT*) and black (*iRT*) genotypes contain proanthocyanidins (PAs) but only black ones contain anthocyanin[41].

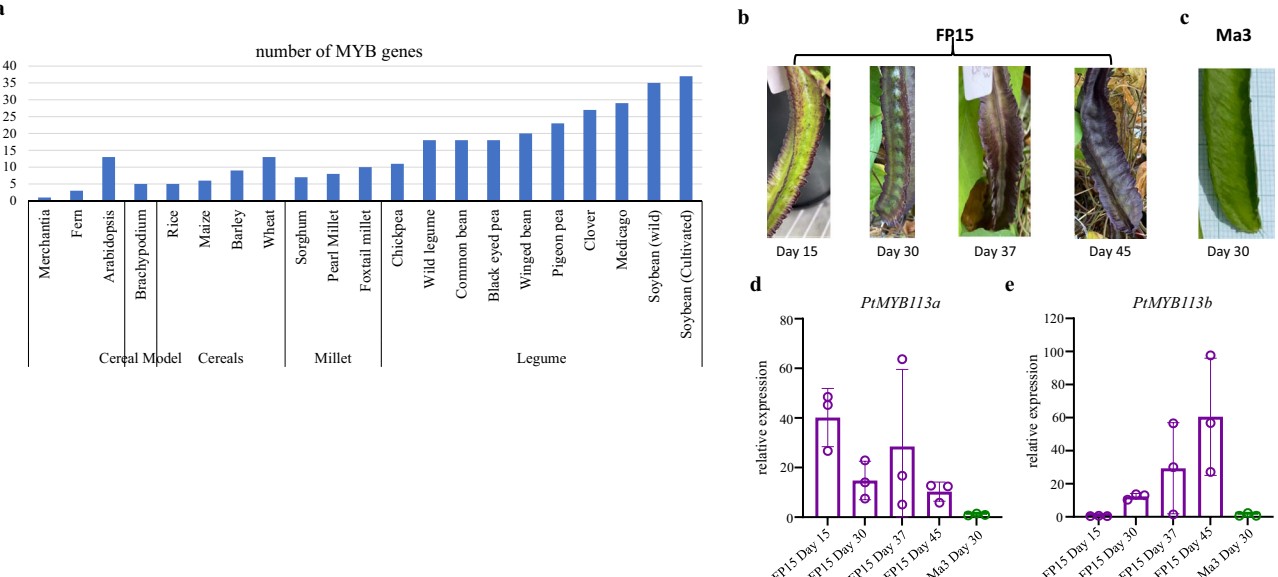

**Fig. 6 | Characterisation of MYB genes involved in the flavonoid biosynthetic pathway. a** Number of genes detected having homeobox-like and SANT/Myb domains in 27 species. Developing pod colour variation observed in FP15 (**b**) comparing to the green pod of Ma3 at Day 30 after anthesis (**c**). The relative expression level (RT-qPCR) of (**d**) *PtMYB113a* and (**e**) *PtMYB113b* in the pods of FP15 from 15, 30, 37 and 45 days after the day of anthesis in comparison with the green pods of Ma3 at Day 30 after anthesis. Data represent mean ± SEM from biological triplicates, *n* = 3. Source data are provided in a Source data file.

Both *P1* and *P2* purple loci are on Pt05, approximately 50 cM apart (Recombination Fraction = 0.32). It is noteworthy that the high LOD scores observed throughout Chr05.1 correlate well with the clustering of genes involved in anthocyanin biosynthesis pathway (Supplementary Fig. 13 and Supplementary Data 14). A search through bidirectional BLAST based on sequence homology and domain structures in the MYB TFs modulating anthocyanin biosynthesis in Arabidopsis[42] has identified 17 MYB genes comprising of R2R3 and R3 repeats in winged bean potentially involved in the regulation of anthocyanin accumulation (Supplementary Data 15). These MYB TFs are present in higher numbers in legumes than selected monocotyledonous species and early land plants (Fig. 6a). Of these, with regards to the Arabidopsis AtMYB113/AtMYB114/AtMYB90 (PAP2; AT1G66370, AT1G66380, AT1G66390, respectively) cluster, a similar tandem triplication is seen in winged bean (*PtMYB113a*: Psote05G0036500, *PtMYB113b*: Psote05G0036600, and *PtMYB113c*: Psote05G0036700) at the purple locus together with the AtMYB75(PAP1) orthologue (named as *PtMYB114*: Psote05G0098700 based on sequence similarity) approximately 9.5 Mb apart (Supplementary Data 14). These tandem replicated genes are also homologous to the *R* locus gene (Glyma.09G235100) in soybean[43] and *VuMYB90-1* in cowpea[44]. From phylogenetic analysis, four MYB TFs belong to the Subgroup 6 (SG6) of R2R3 MYB (which is the anthocyanin-related clade) were identified (Supplementary Fig. 14 and Supplementary Data 16)[45]. Their orthologues are known to be the activators for late biosynthetic genes through interactions at their R3 repeats with a conserved motif ([D/E]L_{x2}[R/K]_{x3}L_{x6}L_{x3}R) in the N-terminal MYB-interacting region (MIR) of the bHLH TFs (Supplementary Data 17)[45,46]. The SG6 KPRPR[S/T]F motif is more conserved in *PtMYB113a*, *PtMYB113b* and *PtMYB113c* than in *PtMYB114* but all have the conserved anthocyanin-promoting (S/A)NDV motif (Supplementary Data 17)[47,48].

Comparing the genotype of the green parent (Ma3) with the purple parent (FP15) a number of sequence changes could be observed. A non-polar neutral glycine is substituted by a hydrophilic negatively charged glutamic acid (Gly195Glu) at Exon 3 of *PtMYB113a* in the purple cultivar (Supplementary Data 18). Despite *PtMYB113a* and *PtMYB113b* being expressed at relatively low abundance (FPKM; Supplementary Data 14) yet they had apparently higher expression than their green counterpart (Fig. 6b, c), this is likely to be sufficient to activate the late biosynthetic genes, as similar expression profiles have been observed in bHLH (TT8/GL3/EGL3) and TTG1 to form the ternary MBW complex. While the sequences after the R3 domain are highly variable among species, the purple genotype is heterozygous for a three amino acid insertion/deletion in Exon 3 of *PtMYB113b* (114DKK, Supplementary Data 17) while the green Ma3 parent is homozygous for the deleted form. As these amino acids occur within the predicted alpha helix region R3, the deletion may affect the efficiency of binding to the complex in the green parent (Supplementary Fig. 15). The effect of the polarity difference also remains to be explored, as the substitution of a hydrophilic uncharged glutamine by a positively charged histidine in the purple genotype is observed (Gln112His) when compared to the green parent. Surprisingly, being a highly inbred line FP15 is heterozygous for three out of these seven non-synonymous variants. In Arabidopsis, the purple pigmentation in 12 day-old seedlings is only observed under high light exposure and phosphorylation at both Thr-126 and Thr-131 of MYB75/PAP1 by MPK4, which prevents its degradation by 26s proteosome, thereby enhancing anthocyanin accumulation[49]. With the lack of threonine in FP15 within that region, the best potential phosphorylation sites in *PtMYB113b* are likely to be at Ser-105 and Ser-113 in the purple line, whereas these would be Thr-105 and Ser-113 in the green cultivar. A recent study has indicated that conformational changes caused by serine phosphorylation are smaller than the on-off switch-like function from threonine phosphorylation[50]. Intriguingly, one of the four copies of *PtMPK4* with moderate expression (2nd highest) is located approximately 3.5 Mb from *PtMYB113b* (Supplementary Data 14). It is of note that the purple locus might be a rare allele as only 9.2% of the World Vegetable Centre collection has purple, brown-black or black seeds. It is evident that structural genes from the anthocyanin pathway, such as *F3'H*, *F3'5'H*, *DFR*, *ANS*, *ATs* and *GTs* were up-regulated to different extents between FP15 and Ma3, following the activation by the MBW ternary complex (Fig. 7 and Supplementary Data 14). The structural variants observed in *Pt5MAT1* and *Pt5MAT2* (anthocyanin acyltransferases (ATs), involved in form modification[51]) as well as their transcript abundance data could suggest these could also be reasonable candidate genes at *P1* locus. These include a missence mutation and four substitutions (Thr22Ser,

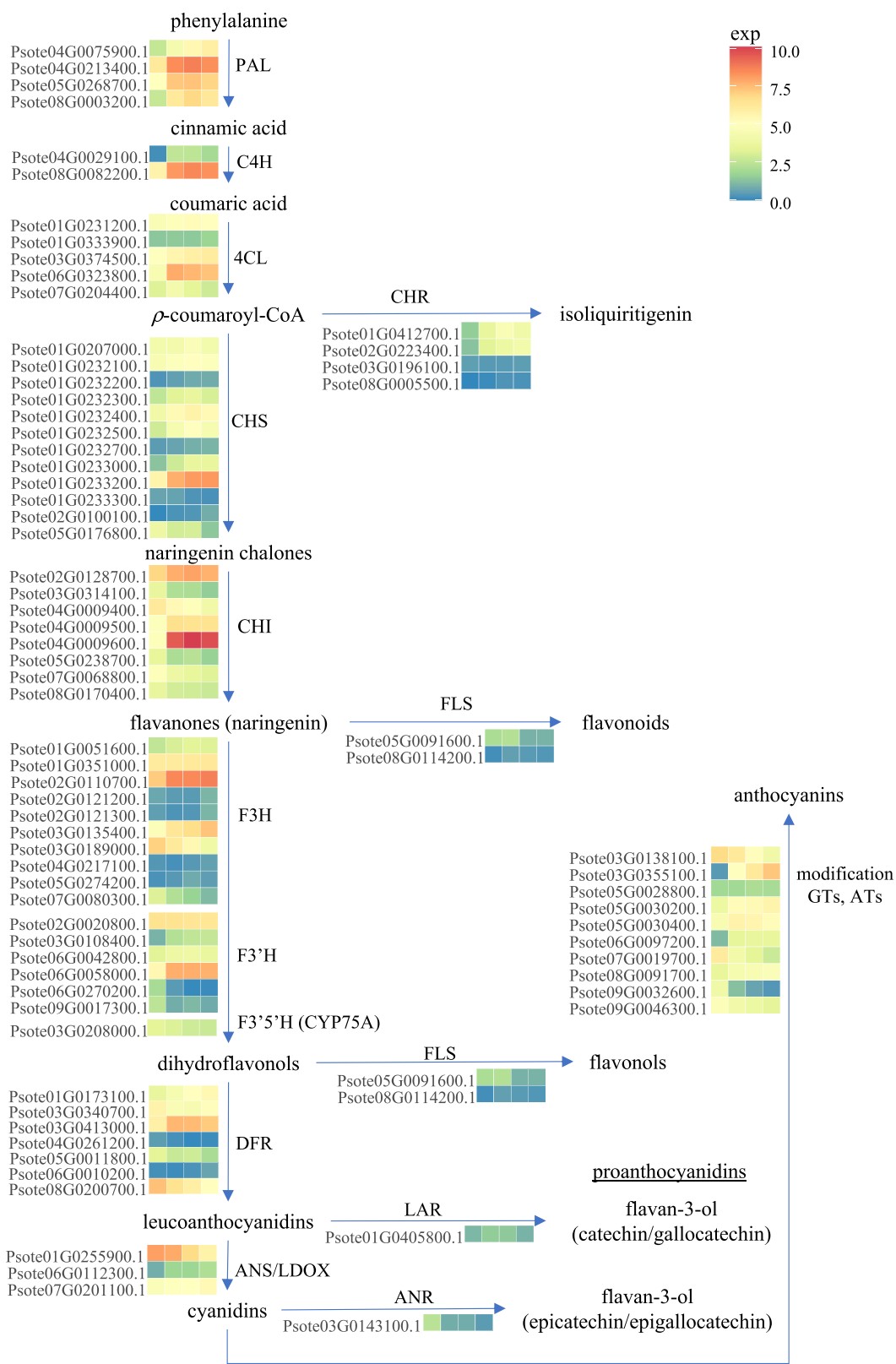

**Fig. 7 | Heatmap of genes involved in the anthocyanin biosynthesis pathway in developing pod.** Transcriptomic profiles of anthocyanin biosynthesis structural genes at Day 15, 30, 37 and 45 after anthesis (from left to right) in the maturing purple pods with the colour scale reflecting $\log_2$(FPKM+1) values. PAL: phenylalanine ammonia lyase, C4H: cinnamate-4-hydroxylase, 4CL: 4-coumarate CoA ligase, CHR: chalcone reductase, CHS: chalcone synthase, CHI: chalcone isomerase, FLS: flavonol synthase, F3H: flavanone-3-hydroxylase, F3'H: flavonoid 3'-hydroxylase, F3'5'H: flavonoid 3'5'-hydroxylase DFR: dihydroflavonol 4-reductase, LAR: leucoanthocyanidin reductase, ANS/LDOX: anthocyanidin synthase/leucoanthocyanidin dioxygenase, ANR: anthocyanidin reductase, GTs: glucosyltransferases; ATs: anthocyanin aromatic acyltransferases. Source data are provided in a Source data file.

Lys272Glu, Ser314Gly and Val447Asp) in *PtMAT1* leading to protein size differences, basic-to-acidic, hydroxyl-to-aliphatic and hydrophobic/non-polar to hydrophilic/polar, respectively, as well as a substitution from a negatively charged glutamic acid to an uncharged glutamine (Glu214Gln) of *Pt5MAT2* (Supplementary Data 18).

Being the candidate gene at the *P2* locus, a Ser63Pro mutation in *PtWRKY70* (Psote05G0080300) has caused hydroxyl-to-aliphatic and polar-to-nonpolar changes among the parental lines despite the WRKY domain not being affected (Supplementary Fig. 16). Its orthologue *AtWRKY70* is better characterised for senescence and defence signal transduction due to its activation by salicylic acid (SA)[52]. Nevertheless, prominent reductions in anthocyanin could be observed in *Atwky70-1* and *Atwrky70-2* mutant lines[53]. In pear, *PbWRKY75* binds to the promoters of *PbDFR*, *PbUFGT* and *PbMYB10b* (but not PbMY10) to enhance their transcriptional activities[54]. The purple FP15 orthologue has a classical 6-bp light responsive cis G-box element 'CACGTG', while Ma3 line has a 'CACGTT' G-box variant. Through transient expression in lima bean cotyledon and stable expression in soybean root hairy roots studies, a change in the first or last nucleotide in the G-box caused significantly lower promoter activity[55]. While this remains to be verified, W-box elements are also predicted *in-silico* in the promoter regions of *PtMYB113a*, *PtMYB113b* and *Pt5MAT1*.

In brief, activation from the MDW regulatory complex at *P1* locus mediating transcriptional activities of anthocyanin structural genes, potentially enhanced by WRKY TF at *P2* locus, leads to the purple pigmentations observed in the calyx, pod, and seed coat of FP15 genotype. The most likely candidate for the lack of pigment observed in Ma3 is the three amino acid deletion observed in the PtMYB113b in the green parent.

## Discussion

The winged bean genome constructed is of value for molecular breeding improvement as most other plant species with sequenced genomes are taxonomically distant, limiting the cross-species use of positional information. Stefanović and colleagues have suggested that *P. tetragonolobus* diverged approximately 11 Mya from *Erythrina soursae*[56]. The common ancestor between this clade and the clade consisting of Phaseoleae subtribe Glycininae and subtribe Phaseolinae which arose approximately 21 Mya, with an estimated divergence slightly earlier (approximately 23 Mya) from the Cajaninae clade[57]. This winged bean genome lays the foundation for crop genome guided breeding improvement programmes, to enhance the wider adoption of the species as a protein source for the future, given the limited conserved synteny with the well annotated *P. vulgaris* genome. The high levels of genetic diversity seen within the breeding material diversity study could be explained by: (i) undocumented early exchange of material (ii) geographical preferences for different edible parts and cropping methods having limited the loss of genetic diversity overall, (iii) seed phenotypes have not been as strongly visually distinctive compared to other legumes, particularly with limited variation in seed mottling and hilum pattern, with the major selection focus on the vegetative parts of the plant.

Thus far, our approach in integrating Illumina, ONT, Bionano and genetic mapping have resulted in a pseudo-chromosome level genome spanning 580.28 Mb. The genome annotation has facilitated the identification of a number of candidate genes associated with QTL relevant to ideotype development for breeding selection, in particular, variability for seed protein content, plant architecture and plant phytochemicals. In addition, it has been possible to identify the regions of the genome containing QTL flanking marker variation, potentially allowing the development of simple KASP markers for further selection of desirable architectural, protein and nutritional traits. Selection for alleles of genes associated with these candidates

(if validated) could subsequently be utilised for varietal development or flanking markers around the QTL of interest could be used directly for breeding.

## Methods

### Library preparation of material for whole genome sequencing

The common paternal line Ma3 for both Cross XT and Cross XB was selected for whole genome sequencing, based on single-seed descent derived material from the individual plant used during controlled hybridisation. The plant was grown in the Future Crop glasshouse under 28 °C day/23 °C night temperature cycle (Sutton Bonington, UK). High molecular weight (HMW) genomic DNA was isolated from 100 mg fresh leaves using DNeasy Mini Kit (Qiagen). DNA quantity and quality were assessed by Nanodrop, Qubit dsDNA BR assay (Thermo-Fisher) and the Genomic DNA ScreenTape assessed (Agilent) prior to library preparation.

For ONT library preparation, a library prep was prepared from 400 ng of DNA (ONT; SQK-RAD004) and run over a MinION flow cell (ONT; FLO-MIN106 R9.4) on the GridION X5 mk. A small-scale ligation sequencing library (ONT; SQK-LSK109) was also prepared and run over a MInION flow cell on the GridION. A large-scale ligation sequencing library was subsequently prepared and the entire library prep was loaded onto a PromethION flow cell (ONT; FLO-PRO001) and sequenced on the PromethION platform.

For Illumina library preparation, an Illumina-compatible sequencing library was prepared from 500 ng of DNA using the KAPA HyperPlus Kit (Roche) and the KAPA Single-Indexed Adaptor Kit, Illumina Platforms, Set B (Roche). Genomic DNA was fragmented for 10 min and 2 cycles of PCR were used during the library amplification stage. Post-amplification SPRI-based size selection was performed using a 0.7/0.9 ratios of AMPure XP beads (Beckman Coulter). The library was quantified using the Qubit Fluorometer and the Qubit dsDNA HS Assay Kit (ThermoFisher) and the fragment length distribution was assessed using the Agilent TapeStation 4200 and the Agilent High Sensitivity D1000 ScreenTape Assay (Agilent). Final library QC was performed using the KAPA Library Quantification Kit for Illumina (Roche). The library was sequenced on the Illumina Next-Seq500 (NextSeq Control Software v2.2.0) using a NextSeq500 Mid Output 300 cycle kit v2.5 (Illumina) to generate 150-bp paired-end reads.

### Bionano mapping

High molecular weight genomic DNA was extracted from fresh young leaves using the Bionano Prep Plant Tissue DNA Isolation Kit (Bionano Genomics) according to the Isolation Base Protocol (Bionano Genomics: Document #30068-RevD). Due to the partially insoluble nature of the DNA, quantification was not possible so 21 μL (including the bulk of DNA) was taken into labelling.

DNA was labelled with the DLE-1 enzyme using the Bionano Prep DLS Kit (Bionano Genomics) following the standard protocol (Bionano Genomics: Document #30206-RevF). The labelled sample was quantified by Qubit Fluorometer 4 (Thermo Fisher) and the Qubit dsDNA HS Assay Kit (Thermo Fisher). The average concentration of the labelled sample was 3.36 ng/μL (CV = 0.09). The labelled DNA reaction was run over one flow cell on a Bionano Saphyr Chip (Bionano Genomics) on the Bionano Saphyr to generate 540 Gbp of data. For analysis, the molecules file was used to generate a de novo assembly using the default Bionano Access settings (Bionano Access: 1.3.0; Bionano Tools: 1.3.8041.8044; Bionano Solve: Solve3.3_10252018; RefAligner: 7915.7989rel; HybridScaffold/SVMerge/VariantAnnotation: 10252018). This assembly was used to generate a hybrid scaffold with the winged bean NGS assembly. The hybrid scaffold was constructed using default settings; conflict resolution was set to 'Resolve Conflicts' for both the Bionano assembly and sequence assembly.

## Genome assembly and TE characterisation

Raw ONT reads were firstly filtered by quality score (<7) and read length (<1k), and then were trimmed, corrected and assembled by Canu (version 2.1.1) and NECAT (v0.0.1 update20200803). Raw assembly was merged further by *quickmerge* to improve contiguity. Assembled contigs were first polished twice by ONT long reads using Racon v.1.5.0 and then further polished by Illumina PE reads using Pilon v1.22[67]. The EDTA pipeline[58] with default settings was used for transposable element characterisation with Wicker et al.'s classification[59]. The Illumina reads were subjected to K-mer analyses using both Jellyfish and GenomeScope 2.0.

## Gene annotation

De novo transcriptome assembly was conducted by Trinity v2.11.0 with previously published RNA-seq data (PRJNA374598)[12]. Raw assembled transcripts were clustered by CD-HIT v4.8.1 and corrected by CAP3. Protein coding regions were identified by TransDecoder v5.7.1. Genome annotation was done by MAKER2 v2.31.10 using transcriptome annotation. The completeness of annotation was estimated using BUSCO v5.4.3 embryophyte odb10. The types of transcription factors were predicted from the Plant Transcription Factor Database (PlantTFDB, http://planttfdb.gao-lab.org/index.php). Collinear blocks were defined using default parameters in MCScanX and visualised using SynVisio (https://synvisio.github.io/#/). Functional annotations including GO, PFAM and KEGG terms were performed using eggNOG-mapper v2. Following that, enrichment analysis of duplicated genes was carried out using *clusterProfiler* v4.2.2. The *p* value was set at <0.01 and adjusted in the GSEA GO and over-representation set of KEGG module using Benjamini-Hochberg method.

## Evolution analysis and divergence time estimation

Ten species were included for the investigation of the evolutionary history of winged bean; *Phaseolus vulgaris* v2.1 (Phytozome), *Vigna unguiculata* v1.2 (Phytozome), *Medicago truncatula* Mt4.0v1 (Phytozome), *Lotus japonicus* v1 (Phytozme), *Glycine soja* v1.1 (Phytozome), *Vitis vinifera* v2.1 (as outgroup, Phytozome), *Cicer arietinum*[57] (Legumepedia), *Cajanus cajan* (Legumepedia), *Trifolium subterraneum*[57] (Legumepedia), *Glycine max Lee* v2[57] (Legumepedia) using OrthoFinder v2.5.4 with default settings except parameters '-M msa -T iqtree'. Orthogroups with the minimum of one gene in all species and not more than 30 genes[57] (*n* = 11,251) were then retained and subjected to CAFE5. The divergence age was calculated using the following formula

$$T = Ks/2\lambda \tag{1}$$

where $\lambda = 8.3 \times 10^{-9}$ as the global mutation rate of legume species[60].

## Genotype-by-sequencing using DArTseq

Genomic DNA was extracted from young leaves using Qiagen DNeasy® 96 Plant Kit and subsequently genotyped by DArTseq™ platform (Diversity Arrays Technology Pty Ltd) in which the DNA was fragmented through *Pst*I-*Mse*I double digestion, followed by adaptor ligation, amplification and then sequenced on Illumina Hiseq 2500. SNP and PAV calling were then carried out using proprietary DArT analytical pipelines.

## Genetic mapping

SNP markers from DArTseq were filtered so as to be polymorphic between parental lines and to have no missing values across the populations for a total of 221 and 183 $F_2$ individuals, respectively in XB and XT populations. Maps were constructed using JoinMap® v5, the first linkage groups (LGs) were obtained through a logarithm of odds (LOD) value of 30 using the "grouping (tree)" function. Initial mapping was then performed using a maximum likelihood (ML) algorithm, in order to deal with the computational load of more than 100 markers in

each LG. After sufficient markers had been removed using iterative reduction of markers with ML, final LGs were generated using regression mapping with default parameters: recombination frequency ≤0.40, LOD value ≥1.0, with goodness-of-fit jump threshold for removal of loci equal to 5.0, and rippling performed after every marker was added into the map. Recombination frequency was converted into map distance (cM) using the Haldane's function and markers were checked for nearest neighbour (NN) fit values and mean $\chi^2$ contribution for the goodness-of-fit within each LG map.

Markers showing significant zygotic segregation distortion (SD) were first removed from initial mapping calculations, and later re-introduced using regression mapping. This was monitored by comparing initial LGs (without SD markers) with newly obtained ones (with SD markers), in order to ensure the marker order between non-distorted markers was consistent. To improve the QTL analysis for *qCC*, *qSC*, and *qCLX*, markers on XB.lg05 were filtered out with a 0.95 threshold and the linkage group was mapped using a maximum likelihood algorithm.

## Genetic diversity and population structure

Germplasm included in these analyses were accessions from MARDI, International Institute of Tropical Agriculture (IITA), World Vegetable Centre (AVRDC), U.S. Department of Agriculture (USDA), East-West Seed (EWS) and a donor (listed in Supplementary Data 7). For the evaluation of true biological replicates in a pre-breeding programmes (using DArT™ pipeline), the DArTSeq SNPs were mapped to the hard masked genome in order to filter out SNP at TE sites. Marker quality control was performed by filtering for biallelic SNP loci with minor allele frequency (MAF) >0.01 and missing value of not more than 10%, resulting in 7543 SNPs. Through Neighbour-joining (NJ) tree (mentioned below), biological replicates were confirmed from the pedigree record if they were (i) sharing the same node in the NJ tree, (ii) node length not more than 0.06 (Supplementary Data 7). For further molecular diversity evaluation (using GATK), raw sequence reads from DArTseq were assessed for quality using FastQC v0.12.1. Adaptor sequences were removed using cutadapt. The reads were then aligned to the genome using Burrows-Wheeler Aligner (BWA) mem v0.7.15 with default parameters. Reads were sorted and duplicates were marked by Picard v1.96 in the BAM alignment files. Single nucleotide polymorphisms (SNPs) were called using GATK's (v4.4.0.0) HaplotypeCaller. Variants were called for each sample individually, producing gVCF files. Joint genotyping of variants across all samples was performed using GATK's GenotypeGVCFs. The resulting VCF file contained genotypes and variant information for all samples. SNP variants were further filtered (excluding biological replicates) using vcftools v0.1.16 with following criteria: --maf 0.01 --max-missing 0.6 --minQ 30 --minDP 3, generating 10,567 SNPs. This set of SNP was also subjected to LD analysis.

The retained missing values from both (i) and (ii) were then imputed using Beagle v5.4 with imputation states=400[61]. The ancestry of each individual was estimated using fastSTRUCTURE with admixture as the ancestry model. fastSTRUCTURE was run using the structure.py script provided with the software. The software was run using multiple choices of $K$ ($K = 1$ to $K = 10$) to investigate the structure of the data. Maximum likelihood was used and genotypes were assigned to a subpopulation. The membership coefficient for clusters was 0.7 or else genotypes were considered as admixed. Principal Coordinate Analysis (PCoA) was performed using TASSEL v5. The Neighbour-joining (NJ) tree was permutated with 5000 bootstrap values using *ape* v5.7, *ade4* v1.7, *adegenet* v2.1.10 and *poppr* v2.9.4. The NJ tree was viewed using iTOL.

The average pairwise divergence among genotypes, which represents the nucleotide diversity per bp, π (pi), expected number of polymorphic sites per nucleotide, θ (theta), observed heterozygosity ($H_o$), expected heterozygosity ($H_e$), unbiased expected heterozygosity

(uH$_e$) and pairwise population differentiation (999 bootstrap) were estimated using TASSEL v5 and *dartR* v2.9.7. The normalised measure of difference between the observed (π) and expected (θ) nucleotide diversity, Tajima's D, was also computed. The population genetic analysis was carried out in *poppr* v 2.9.4. Permutation of Monte-Carlo tests was run at 999.

### Linkage disequilibrium (LD) decay
The correlation coefficients ($r^2$) among pairwise comparisons of loci from SNPs ($p < 0.05$) were calculated using TASSEL v5 with default sliding window setting. The LD decay rate was calculated using an R script written by F. Marroni available at https://fabiomarroni. wordpress.com/2011/08/09/estimate-decay-of-linkage-disequilibrium-with-distance/.

### Protein profiling
Ground dried winged bean seeds were passed through a 0.5 mm sieve (Ultra-Centrifugal Mill ZM 200, Retsch). The samples were stored at −80 °C overnight and were then freeze-dried. The protein content was calculated based on modified a Dumas method using a Protein Analyzer (FlashEA® 1112 N/Protein, Thermo Scientific) with the conversion factor 6.25.

### QTL mapping
F$_2$ individuals of the XB Cross were assessed under field conditions together with the parental lines. They were grown on ridges with trellis (1 m × 2 m) and 1.5 m intra-spacing and 1m inter-spacing at the Field Research Centre of Crops For the Future (Semenyih, Malaysia). Seeds were scarified using sandpaper. The seedlings were then transplanted 14 days after emergence (DAE). Fertiliser (NPK 15:15:15) was applied 4 times during the growing season (estimated to be 15 kg/ha), while pesticide was sprayed approximately every 10 days to prevent pest damage of the main stems (Karate – Syngenta, Switzerland) at 0.5 mL/L concentration.

For the materials used for nutritional profiling, five replicates of each parental genotype were grown in a complete randomised design (CRD) from October 2016 to March 2017 in a sandy loam soil (pH 5.3). The XB F$_2$ lines were assessed in three blocks between June to November 2017 in a complete randomized block design (CRBD). The soil had a sandy loam profile with pH 5.0, while the day/night temperatures recorded were 32 ± 0.9 °C and 23 ± 0.3 °C (on site weather station; DeltaT). Traits measurements from 87 individuals (which were not border plants) were recorded as reported[5]. In addition, the average chlorophyll concentration was obtained from the middle leaflets of the top three fully opened leaves, each measured 3 times at 28 DAE using a SPAD-502 (Konica Minolta). For *qTNoB*, the number of branches along the entire stem were calculated, as opposed to *qNoB* in which represents the number of branches within the first ten nodes.

QTL analyses were conducted using MapQTL v6 using both non-parametric and parametric tests. Each trait was first analysed through a Kruskal-Wallis (KW) test, in order to establish single marker-trait associations (at $p < 0.01$). This was followed by Interval Mapping (IM) analysis with LOD threshold calculated through 10,000 permutation tests, taking the genome-wide (GW) significant threshold at α = 0.05. Only QTLs that were consistent between these two tests were reported. A QTL was considered significant when equal to or above the GW-LOD threshold and explained ≥10% of phenotypic variance (PVE). Multiple-QTL model (MQM) mapping was utilised for seed protein content QTL analysis. All QTL have been mapped onto their corresponding LG using MapChart v2.32, including markers as supporting intervals, using a 2-LOD drop.

### RNA sequencing and gene quantification
FP15 plants were grown in a controlled environment chamber at 26 °C/ 20 °C day/night temperature with 12-h daylength at the University of Nottingham Sutton Bonington Campus. The pod and seed tissues of FP15 were sampled between mid-day and 13.00 at Day 15, Day 30, Day 37 and Day 45 after the day of anthesis from different individual plants, whilst pod tissues of Ma3 were sampled at Day 30. The total RNA was extracted using RNeasy® Plant Mini Kit (Qiagen), according to the manufacturer's instructions. Each stage had biological replicates derived from different plants. Library preparation and transcriptomic sequencing were conducted using Illumina NovaSeq6000 (Novogene, Cambridge). The 150-bp paired end reads (summarised in Supplementary Data 19) were mapped using hisat2 v2.0.5 followed by differential quantification analysis at fragments per kilobase of transcript per million fragments mapped (FPKM) level using featureCounts 1.5.0-p3 and DESeq2 v1.20.0. For RT-qPCR, first-strand cDNA was then synthesised using RevertAid Reverse Transcriptase (Thermo Scientific) and oligo(dT)18 and random hexamer (IDT). Quantitative real time PCR was performed using SensiFast™ SYBR® No-ROX kit (Bioline) with each reaction of 10 μl consisting of 250 nM forward and reverse primer each (except 100 nM each primer for *PtTUB6*) and 12.5 ng cDNA. The PCR conditions were set as 2 min for an initial denaturation, 40 cycles of 95 °C for 5 s, 60 °C (*PtMYB113b*) or 62 °C (*PtMYB113a*) for 10 s, 72° for 10 s (for *PtTUB6* and *PtELFa*, 40 cycles of 95 °C for 5 s, 62 °C for 15 s), followed by melt curve analysis from 95 °C to 62 °C with the rate of 0.5 °C/s. Each sample had technical triplicates. *PtELFa* (Psote09G0074300) and *PtTUB6* (Psote05G0184000) were used as reference genes. The changes in expression were then calculated by normalising to both reference genes[62]. The primers for the qRT-PCR are listed in Supplementary Table 5.

### Phylogenetic analysis of R2R3 MYB genes
To identify MYB-TFs in anthocyanin biosynthesis, 13 MYB genes with known roles in the anthocyanin pathway in Arabidopsis were found to consist of three domains (IPR001005, IPR009057, IPR017930) using the pipeline developed by Mutte and Weijers[63]. Genes having these domains at a bit score ≥100 were classified as putative MYB-TFs involved in anthocyanin biosynthesis. For the MYB subgroup characterisation, the amino acid sequences (sequence accessions listed in Supplementary Data 16) were aligned using MUSCLE in MEGA 11 using default settings. The phylogeny was constructed as a NJ tree with 5000 bootstrap replicates.

### Reporting summary
Further information on research design is available in the Nature Portfolio Reporting Summary linked to this article.

## Data availability
Raw sequencing reads were deposited at the NCBI BioProject ID PRJNA808222 with BioSample ID SAMN26037071 [https://www.ncbi. nlm.nih.gov/sra/?term=PRJNA808222]. Raw reads for DArTseq of diversity panel and re-sequencing of two parental lines were deposited at BioProject ID PRJNA1034490. RNA-sequencing reads of maturing pods were uploaded to GSE246575, The genome assembly and annotation are available at Figshare [https://doi.org/10.6084/m9.figshare. 19196255]. Source data are provided with this paper.

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

## Acknowledgements

This work was supported by Crops For the Future (W.K.H., A.S.T., S.M.), University of Nottingham Malaysia (W.K.H., A.S.T., F.M.), Future Food Beacon (C.M., N.T., R.B.) and Deep Seq (F.S., V.W.), University of Nottingham (N.S., S.M.). We are grateful to East West Seed for their contribution of lines from their breeding programme in the Philippines. We are grateful to Dr Jon Stubberfield for technical assistance and method development. We are also grateful to Yuen Loon Soon for technical assistance in primer design and RT-qPCR and Yuet Tian Chong for the tissue materials.

## Author contributions

Alberto Stenfano Tanzi generated mapping populations, managed the field work and phenotypic measurement, developed genetic maps and performed QTL analysis. Fei Sang assembled the genome, characterised TE and gene annotation, performed *K-mer* and BUSCO analyses. Niraj Shah performed GATK variant calling pipeline and population structure analysis. Christopher Moore generated Bionano mapping and assembly. Niki Tsoutsoura conducted chamber trial, sampling and RNA extraction for RNA-sequencing, performed protein profiling and provided materials for RT-qPCR. Rahul Bhosale performed bidirectional BLASTP and domain-search phylogenetic analysis. Victoria Wright contributed to sequencing integration. Wai Kuan Ho performed diversity analysis, candidate gene identification and expression analysis, reseq data analysis, enrichment analysis, comparative genome analysis, research coordination and wrote the manuscript. Alberto Stenfano Tanzi, Fei Sang, Christopher Moore and Niki Tsoutsoura contributed to the writing of the manuscript. Sean Mayes supervised the scope of this study and revised the manuscript. Festo Massawe edited the manuscript. Sean Mayes and Festo Massawe are the project leaders. All authors read and approved the manuscript.

## Competing interests

The authors declare no competing interests.
