## [Peer Review File · Nature Communications]

A genomic toolkit for winged beanReviewers' Comments:

Reviewer #1:

Remarks to the Author:

The manuscript reports the genome assembly of an accession of the species *Psophocarpus tetragonolobus* (winged bean) and its use to identify QTL in two biparental populations, to assess genetic diversity in representative populations, and to predict candidate genes related to different agronomic traits. The manuscript is clear and I think that the study is interesting and provides novel information to increase the information on evolution of Fabidae cultivated species.

I have some concerns, mainly regarding the bioinformatic analysis. I think that a second round of analysis starting from genome assembly could significantly increase the quality of this work. These are my specific comments and suggestions:

1. Based on the results of BUSCO in which only 84% of conserved genes could be recovered, I am concerned about the per-base quality of the assembly. I think that the assembly contiguity and quality could be improved following a different pipeline. For nanopore reads, I recommend the use of NECAT (<https://www.nature.com/articles/s41467-020-20236-7>) for error correction because NECAT implements specific solutions to the error patterns present in Nanopore reads, some of which I think can not be solved with Illumina polishing because the reads would not map. After error correction, Flye (<https://www.nature.com/articles/s41587-019-0072-8>) can be executed on the corrected reads to perform genome assembly. Flye may be able to increase contiguity and implements an algorithm that helps to resolve some repeat structures. The current annotation of repetitive elements suggest that some repeat structures could be collapsed by the current assembly process. Polishing with Illumina reads can still be performed afterwards to further improve the per-base quality of the assembly. Per-base quality is critical to reach a complete gene annotation, which impacts the completeness of the downstream analysis.

2. Regarding annotation of repetitive elements, I think that the results of repeat masker should be obtained and compared to the results of EDTA. This could help to provide a more complete identification of TEs and possibly make the reported percentage of the genome spanned by these elements more similar to that of other genomes. Please report not only the number of TEs per family, but also the total length spanned by the TEs. Although it is likely that TIR elements are more abundant than LTRs, LTRs tend to be longer and have a larger impact over the genome. Total lengths are important to visualize this effect.

3. The manuscript lacks a more informative overview of the results of the structural and functional gene annotation. Please provide separate statistics on how many gene models and transcripts were identified, length distributions (both gene and CDS lengths) number of genes annotated with GOs, pathways, etc. This is needed to assess the quality of the annotation (which is indirectly related to the quality of the assembly). It is not clear if the distribution shown in figure 2b refers to de-novo assembled transcripts or to annotated transcripts in the genome. The legend says that the x-axis is log scaled but the scale does not look logarithmic and, more importantly, I do not think it should be logarithmic. I would assume that it is bimodal because it includes a large number of alternative transcripts. This can be double checked reporting also the distribution of primary transcript lengths. Being the first species sequenced within the phylogeny, I think it is very important to ensure that the quality of both the assembly and the annotation are as high as possible, because this assembly will probably be used as a reference for future works.

4. About the analysis of the DART-seq data, I encourage the authors to request the raw reads and perform an independent analysis using one of the open-source pipelines for reference-based analysis of GBS data. This way, methods can be improved to clearly explain how the diversity data was generated. I think that the number of reported SNPs is very low, both for the biparental populations and for the diversity panel. This can be an issue related to the analysis pipeline. For the biparental

populations, low numbers can be related to the decision of removing SNPs with missing data. This is a very conservative filter for GBS data, which in the case of the biparental populations can throw away well called SNPs that can be used to link and orient scaffolds.

5. Conversely, I am not sure if the analysis included removal of SNPs within the repetitive structures identified in the assembly. This should be done because genotype calls in these SNPs can falsely increase the reported percentages of heterozygous calls per individual. Please also provide distributions of minor allele frequency and observed heterozygosity for each population. These distributions are important to assess the overall quality and identify possible biases in the variability datasets. Finally, you can identify and remove SNPs with high observed heterozygosity because they are likely to be generated by systematic errors in the genotype calling process.

6. For genotype imputation, please use an LD based algorithm such as that implemented in beagle or NGSEP. LD-based imputation strategies are much more accurate than a numerical imputation particularly for autogamous species that retain larger haplotype blocks, compared to allogamous species.

7. Please provide in the methods more information about the germplasm included in the diversity panel. Are all cultivated accessions or there are wild samples?. Are samples obtained from a germplasm bank?

8. Finally, the authors can analyze the orthologs identified in the comparative genomics analysis to infer divergence times with other families such as Phaseolus, based on analysis of molecular evolution data. The first paragraph of the discussion currently looks disconnected from the results because the results do not provide information on species divergence times, although this information can actually be inferred from a comparative genomics analysis of the assembly against assemblies of relatively close species. See <https://genomebiology.biomedcentral.com/articles/10.1186/s13059-019-1650-2> for details on how this can be done.

Minor comments

1. Table 1: The value of mean read length on ONT reads looks incorrect. I have never seen ONT reads with mean read length of 443Kbp. If this number is true, then a much more contiguous assembly should be possible to generate. The median read length for Bionano also looks weird to me. Please double check all numbers in this table and in the corresponding supplementary table.
2. Lines 138-141. These sentences are not really results and are distracting from the actual result, which is the genome assembly. I think that the goal of this work is not to propose a methodology but to present new resources for plant genomics. I would recommend removing these sentences.
3. Lines 163-164. I think it is not necessary to provide an argument on why genetic diversity of a species should be studied. I would remove the sentence and go straight to the results.
4. Line 167. Please clarify that the sentence refers to percentage of heterozygous calls per sample, to avoid confusions with observed heterozygosity (per SNP).
5. Line 187-188. This can actually be assessed measuring and comparing the genetic diversity within this group against other geographic groups.
6. Figure 3 can take some additional space to be more clear. Instead, please make figure 7 a supplementary file

Supplementary table S1. Titles of the sections are not clear

Supplementary table S3. please report total DNA length spanned by the annotated transposons and the percentage of the genome spanned by the elements of each class.

Reviewer #2:

Remarks to the Author:

In the manuscript "A genomic toolkit for 'the soybean of the tropics' – winged bean (*Psophocarpus tetragonolobus*)", the authors first assembled and annotated a genome for the legume species winged bean, then performed population genetic analyses, and also tried to identify the candidate genes in the QTL regions related to three important agriculture traits.

The results provide a resource and some genetic information for the study of winged bean. However, some of the data generated from this study need to be carefully checked, which will mass up future study if they are inaccuracy/wrong.

1. The quality of the genome, particular the genome annotation, is not so high. Also there might be some problems.

(1) Usually, an evaluation of the quality of genome assembly is mainly based Contig N50, I did not find the data in this study. The scaffold N50 of this genome is 3.8Mb, which is relative low with current sequencing technology.

(2) In this study, 26,370 protein coding genes was annotated. BUSCO analysis suggested that only 84.1% complete conserved genes were recovered, which is also at a relative low level.

(3) Usually, on chromosome arm, gene density is higher and TE density is lower; a contrary pattern on the centromere regions. However, no clear pattern was observed in this study. This need a careful check.

(4) the method/mode for the BUSCO analysis most likely is wrong. The "-m genome" parameter was designed to assess the genome assembly continuously. In addition, some methods were not clearly described, for example, the annotation methods for small nucleolar RNA and tRNA were not provided; the software for de novo annotation and the species which homolog protein sequences coming from were not provided.

(5) In addition, regarding the genome size, it is hard to evaluate if the estimation of this study is correct, because it is much smaller than previous report of 1.22 Gbp or 782 Mb.

2. the population genetic analyses, some results are quite unusual.

(1) In this study, 'the level of heterozygosity ranged from 0.4 to 56.6% with a median value at 6.1%', and heterozygosity of 57 accessions were more than 10%. In addition, the genome sequenced material has around 5.4% heterozygosity detected from the 21,412 gene-enriched SNPs, with 49.9% of DArTseq SNPs located within coding regions.

So many winged bean accessions from this study showed such high heterozygosity. This is very strange. I am not sure if this is real. For example, for the *Camellia sinensis*, a plant of self-incompatibility, the heterozygosity is only 2.8%.

These results need to be carefully checked and validated. I think most likely there are something wrong.

(2) for the population analyses, clustering of the accessions usually exhibited geographic pattern, which means related to their planting areas. However, in this study, the clustering does not correlate well with the country of origin. Although the authors explained that this might reflect seed exchange among genebanks with limited documentation or historical exchange with subsequent collection capturing the secondary collection sites, the results need to be carefully checked. Moreover, for the replicates, 44% (25 out of 57) of these were found not to be highly similar when using molecular markers, which making the results more suspectable.

(3) For the LD analysis part, representative and enough markers are essential for the LD estimation, otherwise it will give false result and conclusion. Using 6,149 SNP is obvious not enough.

3. for the candidate gene investigation for agronomic traits

(1) Although the authors tried to pick up the candidate genes based on researches of homolog genes in other species, a further investigation is needed. For example, the correlation of genetic differences or expression differences with phenotypic changes.

(2) Some conclusions the author made was no supported well with the figure or table they provided. For examples, the PtMYB113 and PtMYB114 was apart distantly in Figure 6a, as well as there is not any subgroup information in Figure 6a and Table S16. It is hard to understand the description in line

404-406 "It is deduced from a phylogenetic study that both belong to the Subgroup 6 which is the anthocyanin-related clade (Fig. 6a and Table S17)54, 55, 56". In fact, the Figure 6a didn't support the previous conclusion that MYB113 and MYB114 belong to a subgroup in my opinion. Another example is that in the lines 431 to 433, the author claimed "It is also worth noting that the clustering of anthocyanin biosynthesis pathway genes correlates well with high LOD scores observed throughout Chr05.1 (Fig S10).", while the Fig S10 only showed the global distribution of the identified anthocyanin biosynthetic genes in Arabidopsis and their 53 orthologues in winged bean. There is no relationship with the high LOD score genes on Chr05.1.

4. Other minor questions:

- (1) Some concepts in sequence and genome assemble field were not correct. For example, although after correcting by Illumina reads, the ONT reads are still reads, they can't be scaffolds (Table 1 and line 82).
- (2) The quality of figures in this manuscript is poor. For example, the multiple sequence alignment result in Figure 7 occupies 8 pages; the Figure S1b and figure S5a did not show names of abscissa and ordinate.
- (3) The figure legend didn't give critical information for understanding the corresponding figure. For example, the figure legend for Figure3 did not show and explain the CP, OT and LT subpopulations.
- (4) Some description is unclear. For example, in line 102, "The Class I/Class II transposable elements (TE) ratio was observed to be 0.81". The "ratio" is confusing, which statistic value it means to, the TE fragment counts, length or the intact TE count, length?

Reviewer #3:

Remarks to the Author:

Winged bean has been termed as 'the soybean for the tropics' with potential to replace soybean in many tropical and semi-tropical countries. This paper firstly reported a de novo assembly a very good chromosome level reference genome of winged bean. The authors also investigated the genetic diversity of 171 worldwide accessions for breeding, together with the first two genetic maps and QTL analysis for genomic regions with 34 desirable agronomic traits for breeding including plant architecture, protein content and seed pigmentation. They also preliminarily indicated the candidate genes controlling the some important QTLs. Overall, I think this paper offer valuable resources for the genomic and genetic analysis of winged bean, and will definitely speed up the candidate gene identification of important traits and molecular breeding. However, some key points need to be further addressed before considering accepting it.

Major points:

1. The 171 accessions worldwide were resequenced by GBS and diversity of this germplasm was analyzed. Unfortunately, they do not conduct the GWAS analysis of those traits including plant architecture, protein content and seed pigmentation. This must be done to add extremely high value of these resources. And the potential QTN from GWAS can be overlapped with the QTL from the cross of XB population.
2. Several candidate genes of QTLs of qSSP-1, qSSP-4, qTNoB-1 and qTNoB-2 were predicted and discussed. But these analyses are very preliminary. Most importantly to me, the sequence variations between the parents, Ma3 and FP15 of the candidate genes should be more deeply analyzed to look the causative mutations for the QTL and possible explanations how the mutations contribute to the traits .

Minor points:

1. Please move the regular Fig 4 to Fig 7 to supplementary. And move Fig S7 to regular Figure.
2. In abstract, the accession number is 136, but in the text and table S6 indicated with 171, please confirm.

3. In the 171 accessions, which ones from wild accessions, landraces, or modern cultivars? Please indicate.
4. In Fig S8, please show the pictures of the seeds to show the morphology differences.
5. In Table S6 and Fig S5, please add the full country names in the note of the table and figure legend.
6. The subpopulation of CP, OT, LT, what are they?

REVIEWER COMMENTS

Reviewer #1 (Remarks to the Author):

The manuscript reports the genome assembly of an accession of the species *Psophocarpus tetragonolobus* (winged bean) and its use to identify QTL in two biparental populations, to assess genetic diversity in representative populations, and to predict candidate genes related to different agronomic traits. The manuscript is clear and I think that the study is interesting and provides novel information to increase the information on evolution of Fabidae cultivated species.

I have some concerns, mainly regarding the bioinformatic analysis. I think that a second round of analysis starting from genome assembly could significantly increase the quality of this work. These are my specific comments and suggestions:

1. Based on the results of BUSCO in which only 84% of conserved genes could be recovered, I am concerned about the per-base quality of the assembly. I think that the assembly contiguity and quality could be improved following a different pipeline. For nanopore reads, I recommend the use of NECAT (<https://www.nature.com/articles/s41467-020-20236-7>) for error correction because NECAT implements specific solutions to the error patterns present in Nanopore reads, some of which I think can not be solved with Illumina polishing because the reads would not map. After error correction, Flye (<https://www.nature.com/articles/s41587-019-0072-8>) can be executed on the corrected reads to perform genome assembly. Flye may be able to increase contiguity and implements an algorithm that helps to resolve some repeat structures. The current annotation of repetitive elements suggest that some repeat structures could be collapsed by the current assembly process. Polishing with Illumina reads can still be performed afterwards to further improve the per-base quality of the assembly. Per-base quality is critical to reach a complete gene annotation, which impacts the completeness of the downstream analysis.

Reply: We thank for the reviewer for pointing our the NECAT assembler for ONT – we have repeated the assembly and analysis based on the new version generated. The summary of the improvement from NECAT.

		NECAT			NECAT
Chr01.1	28,151,882	30,448,284	Chr05.1	23,868,018	28,310,307
Chr01.2	266,674	3,897,633	Chr05.2	11,129,282	13,550,307
Chr01.3	5,089,649	5,269,326	Chr05.3	27,776,552	30,676,283
Chr01.4	2,769,053			62,773,852	72,536,897
Chr01.5	300,539				
Chr01.6	6,932,124	7,909,573	Chr06.1	11,689,154	12,533,122
Chr01.7	15,658,326	17,251,893	Chr06.2	3,009,473	3,618,021
Chr01.8	6,526,732	6,798,887	Chr06.3	6,870,892	8,246,468
Chr01.9	19,358,370	20,408,672	Chr06.4	33,306,251	35,654,499
	85,053,349	91,984,268		54,875,770	60,052,110
Chr02.1	38,637,442	42,435,161	Chr07.1	20,072,704	21,657,062
Chr02.2	586,793	689,220	Chr07.2	22,108,370	23,748,855
Chr02.3	8,634,679	9,747,849		42,181,074	45,405,917
Chr02.4	885,534	1,826,451			
Chr02.5	31,651,608	34,075,877	Chr08.1	17,500,097	18,508,053
	80,396,056	88,774,558	Chr08.2	6,408,012	6,902,238
			Chr08.3	1,407,075	8,294,679
Chr03.1	37,029,132	39,017,878	Chr08.4	7,653,287	
Chr03.2	11,936,123	12,387,269		32,968,471	33,704,970
Chr03.3	893,534	1,658,827			
Chr03.4	589,762	841,008	Chr09.1	4,153,804	4,436,285
Chr03.5	28,233,622	30,731,316	Chr09.2	4,352,480	5,047,007
	78,682,173	84,636,298	Chr09.3	16,101,688	707,157
			Chr09.4		17,314,434
Chr04.1	23,875,316	26,038,758		24,607,972	27,504,883
Chr04.2	26,881,282	28,815,495			
Chr04.3	17,988,146	20,578,218			
	68,744,744	75,432,471			

2. Regarding annotation of repetitive elements, I think that the results of repeat masker should be obtained and compared to the results of EDTA. This could help to provide a more complete identification of TEs and possibly make the reported percentage of the genome spanned by these elements more similar to that of other genomes. Please report not only the number of TEs per family, but also the total length spanned by the TEs. Although it is likely that TIR elements are more abundant than LTRs, LTRs tend to be longer and have a larger impact over the genome. Total lengths are important to visualize this effect.

Reply: We have revised this based on comments from referees and the reassembled genome and summarised in L103-L107

3. The manuscript lacks a more informative overview of the results of the structural and functional gene annotation. Please provide separate statistics on how many gene models and transcripts were identified, length distributions (both gene and CDS lengths) number of genes annotated with GOs, pathways, etc. This is needed to assess the quality of the annotation (which is indirectly related to the quality of the assembly). It is not clear if the distribution shown in figure 2b refers to de-novo assembled transcripts or to annotated transcripts in the genome. The legend says that the x-axis is

log scaled but the scale does not look logarithmic and, more importantly, I do not think it should be logarithmic. I would assume that it is bimodal because it includes a large number of alternative transcripts. This can be double checked reporting also the distribution of primary transcript lengths. Being the first species sequenced within the phylogeny, I think it is very important to ensure that the quality of both the assembly and the annotation are as high as possible, because this assembly will probably be used as a reference for future works.

Reply: We have revised the relevant section based on the reassembly and reviewer comments L109-L122

4. About the analysis of the DART-seq data, I encourage the authors to request the raw reads and perform an independent analysis using one of the open-source pipelines for reference-based analysis of GBS data. This way, methods can be improved to clearly explain how the diversity data was generated. I think that the number of reported SNPs is very low, both for the biparental populations and for the diversity panel. This can be an issue related to the analysis pipeline. For the biparental populations, low numbers can be related to the decision of removing SNPs with missing data. This is a very conservative filter for GBS data, which in the case of the biparental populations can throw away well called SNPs that can be used to link and orient scaffolds.

Reply: The DARTSeq SNP number has reduced from ~20k to ~6k SNP mainly due to the MAF threshold (17,964 SNP with <20% missing value among 168 accessions, SNP = 6714 at MAF > 5%; SNP = 8548 when MAF > 1%). As only a subset of genome was sequenced through DarT Seq (*PstI/MseI* digestion) and with majority of the SNP (89.5%) are from non-TE region (determined after repeat masking), it might not be surprising to obtain less polymorphic markers particularly in an autogamous legume species. We intend to use reseq to improve snp density in the future. For example,

- 1) GWAS study was conducted using 9,825 SNP (Illumina Infinium BARCBear6K_3 BeadChip assay and DARTSeq) among 144 common bean accessions with filtering at 25% missing value and 1% MAF (<https://www.nature.com/articles/s41438-020-00434-6#Sec16>).
- 2) Similarly, 1,568 DARTSeq SNPs were derived from 320 chickpea lines whereas GBS approach using *ApeKI* resulted in 88,845 SNPs (<https://www.nature.com/articles/s41598-018-30027-2>).
- 3) Snp mining from transcripts in pigeonpea (<https://www.nature.com/articles/nbt.2022#Sec2>)

Supplementary Table 16 SNP mining in different crossing parents

Genotype combinations	Number of SNPs identified	Number of genes containing the SNPs
Across 12 genotypes	28,104	17,876
ICPL 20096 × ICPL 332	3,410	2,069
BSMR 736 × TAT 10	2,164	1,003
TTB 7 × ICPL 7035	2,679	1,850
ICPB 2049 × ICPL 99050	3,875	2,472
ICPL 332 × ICPL 7035	3,380	2,132
ICP 28 × ICPL 87091	16,651	11,875

5. Conversely, I am not sure if the analysis included removal of SNPs within the repetitive structures identified in the assembly. This should be done because genotype calls in these SNPs can falsely increase the reported percentages of heterozygous calls per individual. Please also provide

distributions of minor allele frequency and observed heterozygosity for each population. These distributions are important to assess the overall quality and identify possible biases in the variability datasets. Finally, you can identify and remove SNPs with high observed heterozygosity because they are likely to be generated by systematic errors in the genotype calling process.

Reply: Below is the distribution of the het % among the accessions (Table S8). All 7,132 SNPs (based on SNP from the masked genome) are not from repetitive structures. MAF of each population is listed in Table S11. Subpopulation OT has a higher average MAF value than the other two subpopulations.

6. For genotype imputation, please use an LD based algorithm such as that implemented in beagle or NGSEP. LD-based imputation strategies are much more accurate than a numerical imputation particularly for autogamous species that retain larger haplotype blocks, compared to allogamous species.

Reply: We have addressed this. In L259 – L272 and L645-L647

7. Please provide in the methods more information about the germplasm included in the diversity panel. Are all cultivated accessions or there are wild samples?. Are samples obtained from a germplasm bank?

Reply: We have included this information in Table S8.

8. Finally, the authors can analyze the orthologs identified in the comparative genomics analysis to infer divergence times with other families such as Phaseolus, based on analysis of molecular evolution data. The first paragraph of the discussion currently looks disconnected from the results because the results do not provide information on species divergence times, although this information can actually be inferred from a comparative genomics analysis of the assembly against assemblies of relatively close species. See <https://genomebiology.biomedcentral.com/articles/10.1186/s13059-019-1650-2> for details on how this can be done.

Reply: We have addressed this L137-147 and have estimated a date of divergence as suggested.

Minor comments

1. Table 1: The value of mean read length on ONT reads looks incorrect. I have never seen ONT reads with mean read length of 443Kbp. If this number is true, then a much more contiguous assembly should be possible to generate. The median read length for Bionano also looks weird to me. Please double check all numbers in this table and in the corresponding supplementary table.

Checked

2. Lines 138-141. These sentences are not really results and are distracting from the actual result, which is the genome assembly. I think that the goal of this work is not to propose a methodology but to present new resources for plant genomics. I would recommend removing these sentences.

Removed

3. Lines 163-164. I think it is not necessary to provide an argument on why genetic diversity of a species should be studied. I would remove the sentence and go straight to the results.

Removed

4. Line 167. Please clarify that the sentence refers to percentage of heterozygous calls per sample, to avoid confusions with observed heterozygosity (per SNP).

Clarified

5. Line 187-188. This can actually be assessed measuring and comparing the genetic diversity within this group against other geographic groups.

We have made numerous comparisons between the identified groups

6. Figure 3 can take some additional space to be more clear. Instead, please make figure 7 a supplementary file

Done

Supplementary table S1. Titles of the sections are not clear

Supplementary table S3. please report total DNA length spanned by the annotated transposons and the percentage of the genome spanned by the elements of each class.

Reply: We have addressed this in Table S3 and S4 after reassembly and reanalysis.

Reviewer #2 (Remarks to the Author):

In the manuscript "A genomic toolkit for 'the soybean of the tropics' – winged bean (*Psophocarpus tetragonolobus*)", the authors first assembled and annotated a genome for the legume species winged bean, then performed population genetic analyses, and also tried to identify the candidate genes in the QTL regions related to three important agriculture traits.

The results provide a resource and some genetic information for the study of winged bean. However, some of the data generated from this study need to be carefully checked, which will mass up future study if they are inaccuracy/wrong.

1. The quality of the genome, particular the genome annotation, is not so high. Also there might be some problems.

(1) Usually, an evaluation of the quality of genome assembly is mainly based Contig N50, I did not find the data in this study. The scaffold N50 of this genome is 3.8Mb, which is relative low with current sequencing technology.

Reply: The scaffold N50 using ONT and Illumina was at 10.9 Mb, which was later improved to 28.3 Mb with Bionano mapping (summarised in Table 1 and above).

(2) In this study, 26,370 protein coding genes was annotated. BUSCO analysis suggested that only 84.1% complete conserved genes were recovered, which is also at a relative low level.

Reply: This has been improved to 96.2% with the new assembly.

(3) Usually, on chromosome arm, gene density is higher and TE density is lower; a contrary pattern on the centromere regions. However, no clear pattern was observed in this study. This need a careful check.

Reply: This has also been improved with the additional TE pipeline and the revised assembly

(4) the method/mode for the BUSCO analysis most likely is wrong. The “-m genome” parameter was designed to assess the genome assembly continuously. In addition, some methods were not clearly described, for example, the annotation methods for small nucleolar RNA and tRNA were not provided; the software for de novo annotation and the species which homolog protein sequences coming from were not provided.

“-m genome was removed. The annotation completeness was assessed using the annotated gene set and protein mode. small nucleolar RNAs were predicted by snoscan; tRNAs were predicted by tRNAscan-SE. De novo gene annotation was done by SNAP and AUGUSTUS. Homolog protein sequences of soybean (Glycine max v2.1) were used in the initial annotation.”

(5) In addition, regarding the genome size, it is hard to evaluate if the estimation of this study is correct, because it is much smaller than previous report of 1.22 Gbp or 782 Mb. The BUSCO at 96.2% for the reassembly and the fact that only 6.4Mb of scaffold was not incorporated, suggests that the current assembly at least has good representation of the coding regions.

2. the population genetic analyses, some results are quite unusual.

(1) In this study, ‘the level of heterozygosity ranged from 0.4 to 56.6% with a median value at 6.1%’, and heterozygosity of 57 accessions were more than 10%. In addition, the genome sequenced material has around 5.4% heterozygosity detected from the 21,412 gene-enriched SNPs, with 49.9% of DArTseq SNPs located within coding regions.

So many winged bean accessions from this study showed such high heterozygosity. This is very strange. I am not sure if this is real. For example, for the Camellia sinensis, a plant of self-

incompatibility, the heterozygosity is only 2.8%.

These results need to be carefully checked and validated. I think most likely there are something wrong.

Reply: After removing SNP at TE region as recommended by Reviewer #1 (in which 89.48% of the SNP are retained), the heterozygosity of Ma3 genome sequenced material was found to be 9.75% from DArTSeq-SNP whereas estimated to be at 0.399% from K-mer analysis using GenomeScope Profile. As such, this is more likely due to the gene enriched SNP from DArTseq approach. We have discussed at Line 183 -194.

My own experience of using DarT Seq on tea is that it is highly heterozygous. However, it is also clonally propagated (cuttings), so some confusion may arise between analyses which combined multiple samples of the same clone and multiple clones.

(2) for the population analyses, clustering of the accessions usually exhibited geographic pattern, which means related to their planting areas. However, in this study, the clustering does not correlate well with the country of origin. Although the authors explained that this might reflect seed exchange among genebanks with limited documentation or historical exchange with subsequent collection capturing the secondary collection sites, the results need to be carefully checked. Moreover, for the replicates, 44% (25 out of 57) of these were found not to be highly similar when using molecular markers, which making the results more suspectable.

Reply: Our experience in the field is that (unlike controlled environment rooms, where self-pollination is by far the main source of zygotes) there is repeated insect movement into the flowers and where different genotypes are planted together, between flowers in different plants. This may help to explain the long tail of heterozygotes observed. We have examined various approaches to bagging inflorescences for field work, but without success in the Malaysian environment. In another species we work on (Bambara groundnut; *Vigna subterranea*) we do see a clear country patterns and gradient based on geographical distance in sub-Saharan Africa. The current dataset differs clearly from that situation.

(3) For the LD analysis part, representative and enough markers are essential for the LD estimation, otherwise it will give false result and conclusion. Using 6,149 SNP is obvious not enough.

While we appreciate that the number of markers used is relatively limited, how many markers needed partly depends on the breeding system and we have given examples above on assessments which have used similar numbers of markers. The results here suggest that winged bean has a relatively rapid LD decay, indicative of some degree of outcrossing in the history of this material

3. for the candidate gene investigation for agronomic traits

(1) Although the authors tried to pick up the candidate genes based on researches of homolog genes in other species, a further investigation is needed. For example, the correlation of genetic differences or expression differences with phenotypic changes.

We appreciate that this is a provisional analysis and will require further detailed experiment and analysis – our purpose here is to illustrate how the genome can be used to target important traits (protein content, architecture and phytochemicals) for potential breeding work

(2) Some conclusions the author made was no supported well with the figure or table they provided. For examples, the PtMYB113 and PtMYB114 was apart distantly in Figure 6a, as well as there is not any subgroup information in Figure 6a and Table S16. It is hard to understand the description in line 404-406 “It is deduced from a phylogenetic study that both belong to the Subgroup 6 which is the anthocyanin-related clade (Fig. 6a and Table S17)54, 55, 56”. In fact, the Figure 6a didn’t support the previous conclusion that MYB113 and MYB114 belong to a subgroup in my opinion.

Reply to comment: This section has been extensively revised on the basis of the reassembly. The subgroup info in Fig 6a is in the gene label which should have been described better in the figure caption. ‘SG in the gene labels refer to subgroup, antho refers to anthocyanin, PA refers to proanthocyanidin, A refers to activator and R refers to repressor’ added to the figure caption.

Another example is that in the lines 431 to 433, the author claimed “It is also worth noting that the clustering of anthocyanin biosynthesis pathway genes correlates well with high LOD scores observed throughout Chr05.1 (Fig S10).”, while the Fig S10 only showed the global distribution of the identified anthocyanin biosynthetic genes in Arabidopsis and their 53 orthologues in winged bean. There is no relationship with the high LOD score genes on Chr05.1.

Reply: We have revised this, the linking information is at Fig S12 and Table S18.

4. Other minor questions:

(1) Some concepts in sequence and genome assemble field were not correct. For example, although after correcting by Illumina reads, the ONT reads are still reads, they can’t be scaffolds (Table 1 and line 82).

The terminology has been corrected

(2) The quality of figures in this manuscript is poor. For example, the multiple sequence alignment result in Figure 7 occupies 8 pages; the Figure S1b and figure S5a did not show names of abscissa and ordinate.

Figure 7 has been removed to Supplementary data

(3) The figure legend didn’t give critical information for understanding the corresponding figure. For example, the figure legend for Figure3 did not show and explain the CP, OT and LT subpopulations.

The origins of this short hand are given in the text

(4) Some description is unclear. For example, in line 102, “The Class I/Class II transposable elements (TE) ratio was observed to be 0.81”. The “ratio” is confusing, which statistic value it means to, the TE fragment counts, length or the intact TE count, length?

More information given in S3 and S4

Reviewer #3 (Remarks to the Author):

Winged bean has been termed as ‘the soybean for the tropics’ with potential to replace soybean in many tropical and semi-tropical countries. This paper firstly reported a de novo assembly a very good chromosome level reference genome of winged bean. The authors also investigated the genetic diversity of 171 worldwide accessions for breeding, together with the first two genetic maps and QTL analysis for genomic regions with 34 desirable agronomic traits for breeding including plant architecture, protein content and seed pigmentation. They also preliminarily indicated the candidate genes controlling the some important QTLs. Overall, I think this paper offer valuable resources for

the genomic and genetic analysis of winged bean, and will definitely speed up the candidate gene identification of important traits and molecular breeding. However, some key points need to be further addressed before considering accepting it.

Major points:

1. The 171 accessions worldwide were resequenced by GBS and diversity of this germplasm was analyzed. Unfortunately, they do not conduct the GWAS analysis of those traits including plant architecture, protein content and seed pigmentation. This must be done to add extremely high value of these resources. And the potential QTN from GWAS can be overlapped with the QTL from the cross of XB population.

Reply: for seed pigmentation loci, we speculate that it is a rare allele as among 98 accessions phenotyped and hosted in World Vegetable Centre, only 7 accessions have purple, black or brown-black seeds. As such, GWAS might not be suitable. For other traits such as plant architecture and seed protein which are environmental influenced, we do not have resources to phenotype all accessions in a single trial. As the material is from a series of collections, we have yet to generate sufficient seed for a field trial analysis, although seed multiplication is underway.

2. Several candidate genes of QTLs of qSSP-1, qSSP-4, qTNoB-1 and qTNoB-2 were predicted and discussed. But these analyses are very preliminary. Most importantly to me, the sequence variations between the parents, Ma3 and FP15 of the candidate genes should be more deeply analyzed to look the causative mutations for the QTL and possible explanations how the mutations contribute to the traits .

Reply: This has not been possible due to the limitation at the density of GbS SNP in the legume. Re-sequencing of the parental lines has been completed, but further detailed analysis is required to evaluate each of the potential genes. Our purpose at this point is to identify likely candidates, but also use the flanking markers to QTL directly for breeding work.

Minor points:

1. Please move the regular Fig 4 to Fig 7 to supplementary. And move Fig S7 to regular Figure.

At this point, we have moved the MYB tree to supplementary , as the same changes have not been requested by the other reviewers

2. In abstract, the accession number is 136, but in the text and table S6 indicated with 171, please confirm.

Reply to comment: 171 individuals from 136 accessions. Some bio reps in the pre-breeding programme were included to give insight into the out-crossing rate at open field conditions. 171 was reduced to 168 as 4 accessions have unusual high heterozygosity, > 46%, thus excluded.

3. In the 171 accessions, which ones from wild accessions, landraces, or modern cultivars? Please indicate.

Reply: Those are accessions mainly from genebanks, most likely are landraces, we have added the info at the Table S8, in addition to samples supplied by East West Seed from their breeding programme in the Philippines.

4. In Fig S8, please show the pictures of the seeds to show the morphology differences.

Reply: We have added in as Fig 6a.

5. In Table S6 and Fig S5, please add the full country names in the note of the table and figure legend.

Reply: We have addressed that.

6. The subpopulation of CP, OT, LT, what are they?

Reply: It is just to name the sub-population so that it is easier to be referred to.

Reviewers' Comments:

Reviewer #1:

Remarks to the Author:

The manuscript describes the results of different experiments aiming to provide genomic resources for winged bean. This includes a genome assembly, comparative genomics with related species, information of worldwide variability and QTL analysis in a biparental population.

The authors properly addressed most of my concerns in my previous review. I have three final comments about the analysis of genetic diversity and a few minor comments:

1. The authors decided not to reanalyze the raw DArT-seq data, which is unfortunate because, in my experience with DArT-seq, I always obtain about three times the number of SNPs with less missing data and better genotype quality, following open source pipelines. Although probably the results of the analysis would not be invalidated by a reanalysis of the data, the current analysis is not reproducible. The authors should make available the raw data at least of the diversity panel in a public database such as SRA. This would not only improve the reproducibility of the work, but also facilitate other research groups to use the data generated by this paper in future research efforts.

2. Regarding the germplasm diversity analysis, the paragraph from lines 232 to 244 looks speculative. I do not think that statements about selection could be inferred from a simple comparison between expected and observed heterozygosity, especially in an autogamous species. Signatures of selection are usually inferred from WGS data comparing wild and domesticated populations. Coalescence simulations are needed to assess changes in population size. I recommend removing most of this paragraph.

3. The statement in lines 208-210 about reduced diversity in Philippines and India is not well supported by measures of genetic diversity, and moreover, it does not look right. Actually, the expected heterozygosity of the OT and LT subpopulations, which have a high representation in both countries, suggests that the diversity in these countries is actually high, compared to other countries. An actual estimation of genetic diversity within countries is needed if authors wish to compare diversity by country.

Minor comments:

1. Lines 149 to 151. Which control is being used to define enrichment of GO terms?

2. Line 265. Replace "argue" with "suggest"

3. Line 269. It is not clear what is "High LD decay". Actually, the LD decay value of winged bean looks smaller than that of other crops.

4. Line 341. Remove the word "and"

Reviewer #2:

Remarks to the Author:

The revised manuscript was significantly improved. However, some of my concern are not addressed, which are important to evaluate the quality of the work.

(1) For the LD analysis, I strongly suggest the authors to perform the analysis use more markers, or use different sets of markers to make a comparison.

(2) For the candidate gene investigation, more evidences are needed. Current data cannot illustrate how the genome can be used to target important traits for potential breeding work.

Reviewer #3:

Remarks to the Author:

The authors had addressed my concerns. I have no further comments.

REVIEWER COMMENTS

Reviewer #1 (Remarks to the Author):

The manuscript describes the results of different experiments aiming to provide genomic resources for winged bean. This includes a genome assembly, comparative genomics with related species, information of worldwide variability and QTL analysis in a biparental population.

The authors properly addressed most of my concerns in my previous review. I have three final comments about the analysis of genetic diversity and a few minor comments:

1. The authors decided not to reanalyze the raw DArT-seq data, which is unfortunate because, in my experience with DArT-seq, I always obtain about three times the number of SNPs with less missing data and better genotype quality, following open source pipelines. Although probably the results of the analysis would not be invalidated by a reanalysis of the data, the current analysis is not reproducible. The authors should make available the raw data at least of the diversity panel in a public database such as SRA. This would not only improve the reproducibility of the work, but also facilitate other research groups to use the data generated by this paper in future research efforts.

Reply: As requested by the reviewer, we have now repeated all of the relevant analysis for the DArT Seq data using publically available software

ii) Read mapping and variant calling for molecular diversity evaluation (GATK): Raw sequence reads from DArTseq were assessed for quality using FastQC. Low-quality reads and adapter sequences were removed using Trimmomatic with default parameters. The reads were then aligned to the genome using Burrows-Wheeler Aligner (BWA) mem v0.7.15 with default parameters. Reads were sorted and duplicates were marked by Picard v1.96 in the BAM alignment files. Single nucleotide polymorphisms (SNPs) were called using GATK's (v4.4.0.0) HaplotypeCaller. Variants were called for each sample individually, producing gVCF files. Joint genotyping of variants across all samples was performed using GATK's GenotypeGVCFs. The resulting VCF file contained genotypes and variant information for all samples. SNP variants were further filtered (excluding biological replicates) using vcftools v0.1.16 with following criteria: --maf 0.01 --max-missing 0.6 --minQ 30 --minDP 3, generating 10,567 SNPs. This set of SNP was also subjected to LD analysis.

2. Regarding the germplasm diversity analysis, the paragraph from lines 232 to 244 looks speculative. I do not think that statements about selection could be inferred from a simple comparison between expected and observed heterozygosity, especially in an autogamous species. Signatures of selection are usually inferred from WGS data comparing wild and domesticated populations. Coalescence simulations are needed to assess changes in population size. I recommend removing most of this paragraph.

Reply: Based on the new GBS dataset, we have reanalysed conclusions

3. The statement in lines 208-210 about reduced diversity in Philippines and India is not well supported by measures of genetic diversity, and moreover, it does not look right. Actually, the expected heterozygosity of the OT and LT subpopulations, which have a high representation in both countries, suggests that the diversity in these countries is actually high, compared to other countries. An actual estimation of genetic diversity within countries is needed if authors wish to compare diversity by country.

Reply: We have removed this statement. While we have generated additional markers through a reanalysis of the DarT Seq data (10,567), a more specific study may be needed (with preferably resequence data) to make a more definitive statement. At this point, we are not in a position to do this.

Minor comments:

1. Lines 149 to 151. Which control is being used to define enrichment of GO terms?

Reply: It was intended to investigate the GO enrichment in the expanded gene families in the winged bean but this filter step has been overlooked, apologies but corrected in this version.

2. Line 265. Replace "argue" with "suggest"

Reply: done

3. Line 269. It is not clear what is "High LD decay". Actually, the LD decay value of winged bean looks smaller than that of other crops.

Reply: This has been rewritten in light of the new analysis with higher marker numbers and shows a results more consistent with cleistogamy.

4. Line 341. Remove the word "and"

Reply: removed.

Reviewer #2 (Remarks to the Author):

The revised manuscript was significantly improved. However, some of my concern are not addressed, which are important to evaluate the quality of the work.

(1) For the LD analysis, I strongly suggest the authors to perform the analysis use more markers, or use different sets of markers to make a comparison.

Reply: we have reanalysed the raw sequence data generated by DarT Seq to generate higher numbers of markers (10,567) as requested and have used these throughout the relevant analyses.

While this number is limiting for LD, the observed R2 values and derived LD give results in the region expected for a predominantly cleistogamous species.

For the biparental QTL analysis, marker number is unlikely to be as important and the subsequent genome location based analysis examples do provide some initial evidence for relevant genes within the QTL regions identified, supported by expression data. As above, we are not in a position to generate the resequence data as the project is currently unfunded.

Examples of other published studies using limited numbers are given below

- 1) GWAS study was conducted using 9,825 SNP (Illumina Infinium BARCBear6K_3 BeadChip assay and DARTSeq) among 144 common bean accessions with filtering at 25% missing value and 1% MAF (<https://www.nature.com/articles/s41438-020-00434-6#Sec16>).

- 2) Similarly, 1,568 DArTSeq SNPs were derived from 320 chickpea lines whereas GBS approach using *ApeKI* resulted in 88,845 SNPs (<https://www.nature.com/articles/s41598-018-30027-2>).
- 3) SNP mining from transcripts in pigeonpea (<https://www.nature.com/articles/nbt.2022#Sec2>)

Supplementary Table 16 SNP mining in different crossing parents

Genotype combinations	Number of SNPs identified	Number of genes containing the SNPs
Across 12 genotypes	28,104	17,876
ICPL 20096 × ICPL 332	3,410	2,069
BSMR 736 × TAT 10	2,164	1,003
TTB 7 × ICPL 7035	2,679	1,850
ICPB 2049 × ICPL 99050	3,875	2,472
ICPL 332 × ICPL 7035	3,380	2,132
ICP 28 × ICPL 87091	16,651	11,875

(2) For the candidate gene investigation, more evidences are needed. Current data cannot illustrate how the genome can be used to target important traits for potential breeding work.

Reply: The genome has allowed the identification of reasonable candidate genes. Given that most of the agronomically important traits are quantitative and complex, we understand that the candidate gene analysis is a provisional analysis and will require further detailed experiment and analysis. In addition, we have resequenced the parental genotypes for further analysis and added new transcriptomic data from different stages of pod development.

The identification of the expected locations of the genes encoding seed proteins and a comparison with species such as soybean (the closest major genome, from divergence analysis) provide a location-based selection of potential candidates. As discussed in the manuscript, seed protein accumulation is not simple. We have included a transcriptomic analysis of developing pods and seed which provide further guidance on the differentially expressed genes from this subset.

The anthocyanin analysis (MYB113/114) has been further elucidated by resequencing of the parental lines, leading to the identification of a number of clear differences between the MYB113b in green and in purple pods/seeds. In particular, a three amino acid deletion observed at the end of the R3 region in the green pod version may suggest that the expected alpha helix binding of Myb113 to the activation complex is being destabilised. Interestingly, both MYB113a and MYB113b versions in the green parent contain this deletion. The purple parental line (FP15) is heterozygous for MYB113, potentially leading to a slightly distorted 1:1 segregation pattern in maternal tissue. Other potential mutations are present, including a Thr – Ser change which could effect the confirmational change of phosphorylation.

Reviewer #3 (Remarks to the Author):

The authors had addressed my concerns. I have no further comments.

Reply: We thank Reviewer for the comments earlier.

Reviewers' Comments:

Reviewer #1:

Remarks to the Author:

The authors performed data reanalysis and answered my previous concerns. They also made available the DART-seq data. I do not have any further concern and hence I recommend publication of this manuscript. I found a few minor typos that I trust authors can fix during the editorial process:

1. Please double check all figure and table references because I found some figures that do not seem to be referenced accurately. See reference to figure 3 in line 154, figure 6b in line 334 and figure 6c in line 350.
2. Lines 228 and 230. If "Columbian" refers to the country, the spelling is Colombian, not Columbian.
3. Line 288. "... winged bean seeds has ..." it should be "have".
4. Line 545. "... the well annotated *P. vulgaris*". Add the word "genome".
5. Line 699. "The correlation coefficient ..." it should be "coefficients".

Reviewer #2:

Remarks to the Author:

The revised manuscript addressed my questions. I do not have further comments.

REVIEWERS' COMMENTS

Reviewer #1 (Remarks to the Author):

The authors performed data reanalysis and answered my previous concerns. They also made available the DART-seq data. I do not have any further concern and hence I recommend publication of this manuscript. I found a few minor typos that I trust authors can fix during the editorial process:

1. Please double check all figure and table references because I found some figures that do not seem to be referenced accurately. See reference to figure 3 in line 154, figure 6b in line 334 and figure 6c in line 350.

Reply: We thank the reviewer for all the constructive comments, apology to the confusion, we have amended accordingly.

2. Lines 228 and 230. If "Columbian" refers to the country, the spelling is Colombian, not Columbian.

Reply: Thanks for pointing out, edited as suggested.

3. Line 288. "... winged bean seeds has ..." it should be "have".

Reply: Thanks for pointing out, edited as suggested.

4. Line 545. "... the well annotated *P. vulgaris*". Add the word "genome".

Reply: added as suggested.

5. Line 699. "The correlation coefficient ..." it should be "coefficients".

Reply: edited as suggested.

Reviewer #2 (Remarks to the Author):

The revised manuscript addressed my questions. I do not have further comments.

Reply: We thank the reviewer for the previous useful comments.